# Tropospheric Bromine Monoxide Vertical Profiles Retrieved Across the Alaskan Arctic in Springtime

Nathaniel Brockway[1], Peter K. Peterson[2], Katja Bigge[3], Kristian D. Hajny[4], Paul B. Shepson[4,8], Kerri A. Pratt[5], Jose D. Fuentes[6], Tim Starn[7], Robert Kaeser[8], Brian H. Stirm[9], and William R. Simpson[1]

[1]Department of Chemistry and Biochemistry & Geophysical Institute, University of Alaska Fairbanks, Fairbanks, AK, USA
[2]Department of Chemistry, Whittier College, Whittier, CA, USA
[3]Institute of Environmental Physics, Heidelberg University, Heidelberg, Germany
[4]School of Marine and Atmospheric Sciences, Stony Brook University, Stony Brook, NY, USA
[5]Department of Chemistry, University of Michigan, Ann Arbor, Michigan, USA
[6]Department of Meteorology and Atmospheric Science, Pennsylvania State University, University Park, PA, USA
[7]Department of Chemistry, West Chester University, West Chester, PA, USA
[8]Department of Chemistry, Purdue University, West Lafayette, IN, USA
[9]School of Aviation and Transportation Technology, Purdue University, West Lafayette, IN, USA

**Correspondence:** William Simpson (wrsimpson@alaska.edu)

**Abstract.**

Reactive halogen chemistry in the springtime Arctic causes ozone depletion events and alters the rate of pollution processing. There are still many uncertainties regarding this chemistry, including the multiphase recycling of halogens and how sea ice impacts the source strength of reactive bromine. Adding to these uncertainties are the impacts of a rapidly warming Arctic.

We present observations from the CHemistry in the Arctic: Clouds, Halogens, and Aerosols (CHACHA) field campaign based out of Utqiaġvik, Alaska from mid-February to mid-April of 2022 to provide information on the vertical distribution of bromine monoxide (BrO), which is a tracer for reactive bromine chemistry. Data was gathered using the Heidelberg Airborne Imaging DOAS Instrument (HAIDI) on the Purdue University Airborne Laboratory for Atmospheric Research (ALAR) and employing a unique sampling technique of vertically profiling the lower atmosphere with the aircraft via "porpoising" ma-

neuvers. Observations from HAIDI were coupled with radiative transfer model calculations to retrieve mixing ratio profiles throughout the lower atmosphere (below 1000 m), with unprecedented vertical resolution (50 m) and total information gathered (average of 17.5 degrees of freedom) for this region.

A cluster analysis was used to categorize 245 retrieved BrO mixing ratio vertical profiles into four common profile shapes. We often found the highest BrO mixing ratios at the Earth's surface with a mean of nearly 30 pmol mol$^{-1}$ in the lowest

50 m, indicating an important role for multiphase chemistry on the snowpack in reactive bromine production. Most lofted BrO profiles corresponded with an aerosol profile that peaked at the same altitude (225 m above the ground), suggesting that BrO was maintained due to heterogeneous reactions on particle surfaces aloft during these profiles. A majority, 11 of 15, of the identified lofted BrO profiles occurred on a single day, March 19, 2022, over an area covering more than 24,000 km$^2$, indicating that this was a large scale lofted BrO event.

The clustered BrO mixing ratio profiles should be particularly useful for some MAX-DOAS studies, where a priori BrO profiles and their uncertainties, used in optimal estimation inversion algorithms, are not often based on previous observations. Future MAX-DOAS studies (and past reanalyses) could rely on the profiles provided in this work to improve BrO retrievals.

**Key points:**

– BrO profiles retrieved in mostly clear-sky conditions peak at the Earth's surface, consistent with the snowpack as a
source/recycling mechanism for reactive bromine.

– Lofted BrO profile cases were largely found on a single day and corresponded with a particle extinction profile that also peaked above the snow surface.

– Aircraft porpoising, which could be applied to smaller airborne platforms, greatly increases the retrieved vertical information compared to ground-based MAX-DOAS observations.

**Plain Language Summary:**

Reactive bromine is a large driver for the overall air quality of the springtime Arctic because it destroys ozone near the Earth's surface and acts as an oxidant. Ozone is the primary source of OH radicals, which are usual scrubbers of air pollution. During these periods of ozone depletion, reactive halogens, i.e. chlorine and bromine radicals, act to remove air pollution. Despite the importance of reactive bromine to the air quality of the springtime Arctic, there is still uncertainty around its
sources and recycling.

Polar regions are also warming more quickly than other areas on Earth due to climate change, and this is expected to impact bromine chemistry. However, we do not fully understand how warmer temperatures and other associated changes, like loss of sea ice, affect atmospheric bromine chemistry. It is important to observe reactive bromine chemistry in this environment to answer some of these questions.

In this study, we observed the amount of bromine monoxide (BrO), a marker for bromine/ozone chemistry, as a function of altitude across the North Slope of Alaska. We found that BrO was often highest near the ground, confirming the ability of snow to produce and recycle reactive bromine. We identified four common distributions of BrO as a function of altitude, which helps us better understand the many factors that influence this chemistry. The results of this study will help us model the chemistry of the springtime Arctic more accurately in the future.

# 1 Introduction

Reactive bromine in the springtime Arctic boundary layer is a primary driver of ozone depletion events (ODEs) (Simpson et al., 2007b; Wang et al., 2019). These events are particularly important for local and regional air pollution, as they alter the oxidative capacity of the atmosphere leading to faster oxidation of many trace gas pollutants (e.g. hydrocarbons) (Jobson et al., 1994; Cavender et al., 2008; Gilman et al., 2010; Saiz-Lopez and von Glasow, 2012). Much work has been done to understand reactive bromine sources and chemistry, but there are still many uncertainties regarding the multiphase reactions that act as a source of reactive bromine (Pratt et al., 2013; Custard et al., 2017; Jones et al., 2009). Combined with these uncertainties is the influence of a rapidly warming Arctic (Previdi et al., 2021; Rantanen et al., 2022). Further, the North Slope of Alaska, and other Arctic locations, experiences fossil fuel extraction, and the associated anthropogenic emissions influence the halogen chemistry of the area, with the overall impact still unknown (Custard et al., 2015; McNamara et al., 2019).

The gas phase reaction of ozone with bromine atoms causes ODEs, with the subsequent recycling of BrO back to atomic bromine largely occurring through multiphase chemistry that leads to catalytic ozone loss (Simpson et al., 2007b; Wang et al., 2019). BrO may react with $HO_2$ or $NO_2$ to form species that are slow-reacting in the gas phase (HOBr and $BrONO_2$, Thorn et al., 1993; Abbatt, 1994). However, these reservoir species can react on saline surface to produce $Br_2$, which quickly photolyzes in the daytime to produce atomic bromine (Fan and Jacob, 1992; McConnell et al., 1992; Pratt et al., 2013; Wang et al., 2019). This photochemical cycle describes the typical "bromine explosion" (Wennberg, 1999), although there are many more reactions involved in this chemistry, and a more comprehensive explanation can be found in reviews such as Abbatt et al. (2012) or Simpson et al. (2015).

As the oxidative capacity of the atmosphere becomes driven by halogens, the rate of pollution processing is impacted, often with increased oxidation of hydrocarbons, largely by chlorine (Gilman et al., 2010; Hornbrook et al., 2016). Similarly, atmospheric mercury deposition has been observed during ODEs (Schroeder et al., 1998; Steffen et al., 2008; Wang et al., 2019) and occurs in large part via Br atom reactions with gaseous elemental mercury (Stephens et al., 2012; Wang et al., 2019), resulting in oxidized mercury that subsequently deposits, thus increasing the bioavailability of mercury in Arctic regions (Scott, 2001; Brooks et al., 2006).

This chemistry is occurring in a changing Arctic that is warming particularly quickly (Previdi et al., 2021; Rantanen et al., 2022), which can have many impacts on springtime reactive bromine chemistry. For instance, reactive bromine chemistry stops upon snow melt (Burd et al., 2017; Jeong et al., 2022), the timing of which is impacted by warming. $Br_2$ production occurs from the saline snowpack on sea ice and the local tundra (Pratt et al., 2013; Custard et al., 2015), and the distribution of sea ice is clearly impacted by warming. Further, Simpson et al. (2007a) found a positive correlation between BrO (a tracer for reactive bromine chemistry) amounts and air mass contact with first-year sea ice areas. Krnavek et al. (2012) observed bromide depletion in snow on sea ice due to bromine activation to the gas phase, and Peterson et al. (2019) found bromide enrichment in first-year sea ice regions compared to multi-year ice regions, suggesting that the magnitude of the reactive bromine source from saline snowpack on the sea ice could be impacted by the age of the sea ice. As multi-year ice is decreasing with climate change (Kwok, 2018; Howell et al., 2022), this trend could have implications on reactive bromine emissions (Pratt, 2019). As

reactive bromine concentrations are also impacted by atmospheric dynamics (Peterson et al., 2015; Swanson et al., 2020), any changes to Arctic atmospheric dynamics due to warming could also impact halogen chemistry. Further, climate change may also impact the relative importance of different halogen species to surface ozone destruction depending on the location in the Arctic (Benavent et al., 2022).

Prudhoe Bay and the North Slope of Alaska oil fields are located roughly 320 km east of Utqiaġvik, and there are many anthropogenic emissions associated with the oil extraction (Jaffe et al., 1995; Brooks et al., 1997; Floerchinger et al., 2019). Nitrogen oxide ($NO_x$) emissions from the North Slope of Alaska oil fields directly interact with bromine chemistry, altering BrO concentrations in the near-field and impacting the recycling of reactive bromine (Custard et al., 2015; McNamara et al., 2019). Possible Arctic development due to melting sea ice could also increase $NO_x$ emissions in the region in the future (Peters et al., 2011).

Considering these changes to the Arctic environment that are likely already impacting tropospheric reactive bromine and pollution processing in the Arctic, it is important to study the underlying chemical and physical processes. There have been several instances of ground based observations of BrO in the Arctic via in-situ instrumentation (Liao et al., 2011; Peterson et al., 2015), multi-axis differential optical absorption spectroscopy (MAX-DOAS, Tuckermann et al., 1997; McElroy et al., 1999; Carlson et al., 2010; Frieß et al., 2011; Simpson et al., 2017; Benavent et al., 2022; Zilker et al., 2023), and long-path DOAS (e.g. Hönninger et al., 2004; Liao et al., 2011; Stutz et al., 2011). However, ground-based instrumentation is limited to a single location and provides limited information on the vertical structure of trace gases. MAX-DOAS observations can be used to retrieve vertical profiles through the use of optimal estimation (Frieß et al., 2011); however, the resulting profiles often have low vertical resolution, with the ability to resolve only two altitude-dependent values.

Chemical modelling can be particularly useful to study reactive bromine source mechanisms along with specific multiphase reactions (e.g. von Glasow et al., 2002; Marelle et al., 2021; Ahmed et al., 2022; Swanson et al., 2022; Wang and Pratt, 2017). However, as noted in Wang and Pratt (2017), there are few vertically (and horizontally) resolved observations that can be used to evaluate the modelling of different halogen compounds, making observations that can fill this knowledge gap particularly important.

The BRomine, Ozone, and Mercury EXperiment (BROMEX) took place in Utqiaġvik Alaska (formerly Barrow) in spring of 2012, in part to provide more information on the vertical and horizontal structure of reactive bromine in the springtime Arctic, largely through measurement of BrO with an airborne MAX-DOAS (AMAX-DOAS) instrument (General et al., 2014), the same instrument is used in this study. Observations from this instrument provided useful information on how the vertical distribution of BrO is impacted by enhanced convection near sea ice leads, cracks in the sea ice that expose open water (Peterson et al., 2016). That field campaign also observed one instance of increased BrO lofted above the Earth's surface and maintained via multiphase reactions on particle surfaces (Peterson et al., 2017). BrO amounts were also found to be enhanced over inland regions, indicating the role of inland snowpack on bromine recycling (Pratt et al., 2013; Peterson et al., 2018). Additionally, this AMAX-DOAS instrument was used to study the interaction between BrO and fresh $NO_x$ emissions (Custard et al., 2015). The BROMEX field campaign also included three ground-based MAX-DOAS instruments, one located in Utqiaġvik and the other two located on the sea ice (Peterson et al., 2015; Simpson et al., 2017). These studies observed many instances of BrO

lofted above the Earth's surface, as well as a high correlation between observations from the three platforms, indicating that there was little change in BrO over the length scale between the instruments (roughly 30 km). One MAX-DOAS instrument traveled downwind across an open lead and found no BrO enhancement downwind of the lead. Outside of the BROMEX field campaign, Bognar et al. (2020) also observed a correlation of lofted reactive bromine with aerosol particles and attributed this relationship to coarse-mode sea-salt aerosol. Over the past few decades, BrO column amounts also appear to be increasing, possibly due to an increased distribution of first-year sea ice to multi-year sea ice (Bougoudis et al., 2020).

However, despite our improved understanding of Arctic reactive bromine chemistry resulting from the BROMEX field campaign and many other studies, there are still few vertically-resolved observations of reactive bromine, as the ground-based MAX-DOAS studies were only able to retrieve two vertically-resolved values (Peterson et al., 2015; Simpson et al., 2017). The success of the BROMEX field campaign, along with the remaining uncertainties of Arctic reactive bromine chemistry, motivates future airborne observations in and around Utqiaġvik and the research shown in this study.

The data used in this work was collected during the CHemistry in the Arctic: Clouds, Halogens, and Aerosols (CHACHA) field campaign, which was based out of Utqiaġvik, Alaska from mid-February to mid-April of 2022. CHACHA was a multi-disciplinary campaign to study halogen chemistry in the Arctic and the influences of open-lead convection, cloud processing, and anthropogenic emissions. This study utilized the Heidelberg Airborne Imaging DOAS Instrument (HAIDI, General et al., 2014) onboard the Purdue University Airborne Laboratory for Atmospheric Research (ALAR). In this study, we aim to quantify the vertical profile of BrO by utilizing remote sensing with a light aircraft. We provide vertical profiles of BrO (below 1000 m altitude) with 50 m resolution measured via aircraft near Utqiaġvik from late-February to mid-April of 2022.

## 2 Methods

### 2.1 CHACHA Campaign and Location

The CHACHA field campaign took place from mid-February to mid-April of 2022 and was based out of Utqiaġvik, Alaska, which is located at the northernmost point of Alaska between the Chukchi and Beaufort seas. A ground-based station was set up to study turbulence at the Earth's surface and monitor ozone mixing ratios throughout the campaign. Two aircraft were used for the campaign, a University of Wyoming King-Air aircraft and the Purdue ALAR aircraft, which was equipped with instrumentation to study gas-phase and particulate halogens, particle sizes, and cloud droplets. However, this paper only utilizes data from ALAR, which was equipped with a best air turbulence (BAT) probe that provided 50 Hz wind and pressure measurements (Garman et al., 2006, 2008), two roughly 2 Hz temperature probes, a 2B Technologies model 205 ozone monitor with 10 s time resolution, and a Grimm particle counter (model 1.109, briefly described in Peterson et al., 2017) that recorded size-resolved distributions every 6 seconds. Utqiaġvik is also the location of an Aeronet station located at the North Slope of Alaska Atmospheric Radiation Measurement (ARM) facility. This data was critical to characterize the impacts of particle light scattering on our remote sensing observations. Among the numerous other instruments housed at the ARM site is a total sky imager with data on total cloud cover (Flynn and Morris, 2015), which can provide useful context for the conditions under which observations were made.

As Utqiaġvik is located at the northern tip of Alaska, ALAR flew over sea ice to the west, north, and east and over tundra to the southwest, south, and southeast. Few flights went further south than Atqasuk, Alaska, where the topography starts to rise, so the ground elevation was close to sea level for most observations. Regardless, all altitude plots shown throughout this work are above ground level (AGL). The town of Deadhorse, Alaska, which sits in the middle of the Prudhoe Bay oil field, is located roughly 320 km southeast of Utqiaġvik, and there are many facilities in the area that emit pollutants. $NO_x$ emissions in the area were dominated by two specific facilities (the Central Compressor Plant and Central Gas Facility) during the campaign as observed with the HAIDI nadir spectrometer. As these facilities are in close proximity with less than 1 km apart, Prudhoe Bay will be shown as a single point source centred between these facilities throughout this study.

## 2.2 HAIDI and dSCD Retrieval

In this study, HAIDI was configured with two spectrometers having CCD detectors that measure light from either a nadir whisk-broom scanner or a forward near-limb push-broom imager (detailed in General et al., 2014). Both spectrometers recorded incoming light from roughly 301-408 nm allowing for the observation of $NO_2$, BrO, and $O_4$, which is the collision induced absorption of $O_2$. This study only uses observations from the near-limb spectrometer, which recorded incoming light over the 256 CCD channels that were binned into four different elevation angles of -3.6°, -1.9°, 0°, +1.6° relative to the aircraft central axis, each with a field of view of 3.3° full width at half maximum as measured in the field, though the most upward and downward views had some truncation at the edges. The telescopes were fixed with respect to the aircraft, thus the viewing angle varied with the aircraft pitch angle.

HAIDI utilizes the differential optical absorption spectroscopy (DOAS) technique, detailed in Platt and Stutz (2008). Individual trace gases are quantified by identifying their narrow band absorption in measured scattered solar spectra. The logarithm of recorded spectra relative to a reference spectrum, taken in-flight with minimum background trace gas concentrations (more information in Supplement), are fitted with a linear combination of trace gas absorption cross-sections and a polynomial that accounts for the spectrally broader features of Rayleigh and Mie scattering (see DOAS book for details, Platt and Stutz, 2008). The specific fit parameters and cross-section references used to retrieve BrO, $NO_2$, and $O_4$ dSCDs and an example fit retrieval are shown in the supplement (Table S1, Fig. S1).

The coefficients of the DOAS fit routine are differential slant column densities (dSCDs). A slant column density (SCD) describes the total number of trace gas molecules along light paths that lead from the sun through atmospheric light scattering events to the observing telescope. The measurement is differential because the reference spectrum has some amount of trace gas absorption in it, and the dSCD represents the difference in column abundance from this reference. The observed dSCD is thus a function of the amount of trace gas in the atmosphere, location of the trace gas, solar geometry, measurement geometry, and aerosol light scattering. A radiative transfer model is used to account for the solar and measurement geometry, especially for near-limb observations where the sensitivity to different levels of the atmosphere varies by orders of magnitude.

The utility of an AMAX-DOAS instrument is that the combination of observations with different viewing geometries provides information on the vertical structure of the detected trace gas around the flight altitude, as these different viewing geometries provide variable sensitivity to different parts of the atmosphere. For instance, a limb observation (tangent to the Earth)

is most impacted by trace gas absorption at the flight altitude, whereas an upward viewing angle will have higher sensitivity to the atmosphere above the aircraft, and vice versa. Radiative transfer model results describe how trace gases at each altitude impact the observations to determine a corresponding trace gas vertical profile that fits the observed dSCDs.

## 2.3 Porpoising

185 As each individual observation is most sensitive to the area around the flight altitude, we combined observations at different altitudes to increase the amount of information we could retrieve on the vertical trace gas profile, as demonstrated in Baidar et al. (2013). To maximize our retrieved information content, the aircraft often maintained a constant heading while varying altitude between a minimum altitude (usually < 100 m) and a maximum altitude (> 600 m), as seen in Fig. 1 in a method known as "porpoising" (Gerber et al., 2013). By maintaining the heading, the radiative transfer complications of comparing

190 multiple observations is greatly simplified, as the solar geometry is relatively unchanged. We then combined all observations from a porpoise (down and up) to retrieve a single high-resolution BrO and $NO_2$ profile. The mean horizontal distance that the aircraft travelled during these profiles was roughly 35 km. We therefore sacrifice horizontal resolution for vertical resolution. This choice likely has little impact for species that do not exhibit small scale horizontal features, for example when quantifying background BrO profiles which should be relatively constant over such length scales (Simpson et al., 2017). However, this

195 method would be limited in quantifying gases with sharp spatial gradients, such as near-source power plant $NO_2$ plumes, and other gases affected around such sharp gradients.

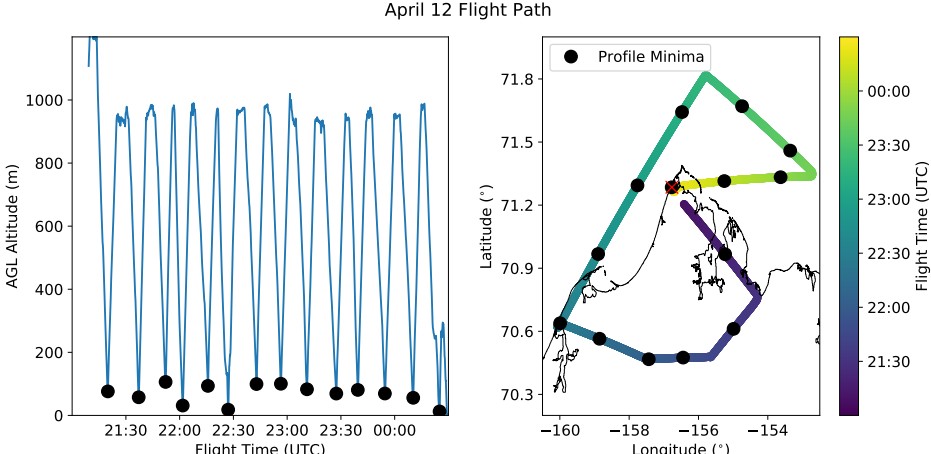

**Figure 1.** Example flight pattern for ALAR during the CHACHA campaign with the (above ground level - AGL) altitude profile shown on the left and the flight map shown as a function of time on the right. The black dots denote the location at the center of each porpoise. By changing altitude and maintaining heading, the radiative transfer considerations are simplified, allowing for easier and more accurate retrieval of high-resolution profiles of BrO and $NO_2$ mixing ratios.

## 2.4 Optimal Estimation

An optimal estimation algorithm is used to determine mixing ratio profiles from AMAX-DOAS observations based on the sensitivity of the measured parameter (dSCD) to the state parameter we wish to retrieve (mixing ratio).

The details of the optimal estimation, or inversion, algorithms used in this study are described in more detail in the supplement, including the standard mathematics from (Rodgers, 2000). These algorithms require radiative transfer modelling to describe how light reaches the four telescopes and thus how gases at different altitudes impact the observed dSCDs for the different viewing angles.

For this work, we used VLIDORT version 2.8 (Spurr et al., 2022), which is a 1-D radiative transfer model that calculates single scattering processes in a full-spherical geometry and multiple scattering processes with a first-order quasi-spherical approximation. The VLIDORT atmosphere used in this work was manually initialized and consisted of 78 layers up to 48 km altitude, with variable grid spacing ranging from 50 m for altitudes below 2 km to 4 km spacing at the top of the grid. The rest of the model initialization is described in the supplement. VLIDORT is used to calculate trace gas box air mass factors (BAMFs, Wagner et al., 2007), which describe the horizontal path-length enhancement of light through each modelled altitude grid cell (Hönninger et al., 2004) and are used as the forward model matrix ($\mathbf{K}$) row vectors for the BrO profile optimal estimation (Baidar et al., 2013).

However, in order to accurately model how light reaches the telescope, it is particularly important to properly represent particle extinction in the atmosphere, as this greatly impacts how light travels through the lower atmosphere. The aerosol optical properties are calculated with a size-resolved bulk Mie code coupled to VLIDORT, with the input parameters described in the supplement, and the particle vertical profile is retrieved using $O_4$ dSCD observations (Frieß et al., 2019). The vertical profile of $O_4$ depends on the square of $O_2$ concentration, and thus on temperature and pressure, meaning that the vertical profile of $O_4$ is relatively well known. The observed $O_4$ dSCDs are reliant on the vertical profile of $O_4$ concentration and how light travels through the atmosphere. In a cloud free atmosphere this is mainly a function of aerosol particle extinction, so these dSCD observations can be used to retrieve the vertical profile of the aerosol extinction coefficient.

For the $O_4$ optimal estimation inversion to determine the particle extinction profile, we utilized the Levenberg-Marquardt method (Levenberg, 1944; Marquardt, 1963), which is an iterative damped least-squares optimization method particularly useful for solving non-linear inverse problems. The particle profile optimal estimation requires VLIDORT to calculate the impact of particle extinction on incoming light with and without $O_4$ absorption (Eq. (7) in the supplement). The details of the optimal estimation algorithm are described in the supplement.

To save computational time, we randomly selected 200 observations, roughly 5% of an average 4,141 observations per flight, from the different porpoises throughout each flight for use in this retrieval of the flight's average aerosol particle profile. Therefore, a constant particle extinction profile is assumed for each flight, as the mostly clear-sky nature of the days when ALAR flew is associated with strong surface temperature inversions that result in a stable atmosphere. These 800 data points, due to the four forward viewing angles, result in excellent particle profile retrievals, with a mean 15 degrees of freedom (DOF). An example of the particle profile averaging kernel matrix and retrieved particle extinction is shown in Fig. 2, where the

averaging kernel represents the fraction of the retrieved information at each altitude that is due to observations at all levels. Measured and modelled $O_4$ dSCDs can be seen in Fig. S3, where modelled dSCDs fit all porpoise observations with a $R^2$ of 0.824.

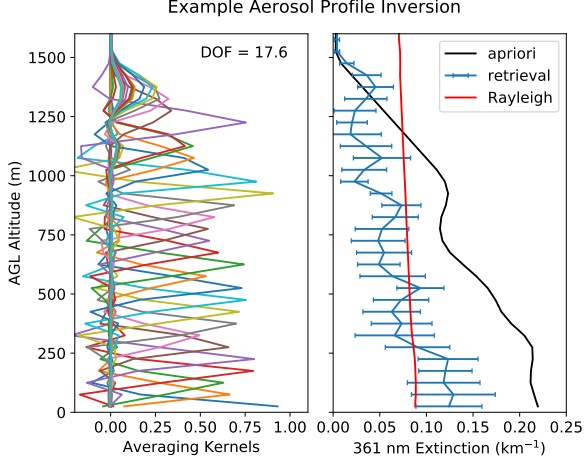

**Figure 2.** Example retrieval of the average particle extinction profile from March 30 with the averaging kernels on the left (DOF of 17.6) and the extinction profile on the right. The a priori extinction is based on the profile shape from the Grimm particle number density from throughout the flight scaled to Aeronet observations at the ARM facility. VLIDORT modelled how light at 361 nm reached our telescopes to perform the $O_4$ inversion, and thus retrieves the 361 nm particle extinction. VLIDORT uses a bulk Mie code to determine aerosol optical properties at other wavelengths. The impact of Rayleigh scattering is shown in red, indicating that the particle and molecular extinction are comparable at the surface for this day.

The retrieval of a BrO profile is more straightforward than for $O_4$ and requires a simple linear inversion that is only performed once per profile. VLIDORT was constrained for each individual flight with the particle extinction profile calculated from the $O_4$ inversion and operated at a wavelength of 350 nm to calculate BAMFs for each flight for all four forward viewing angles by adding the mean aircraft pitch angle to the relative viewing angles for each individual observation. All BrO dSCDs from the four viewing angles are combined for each individual porpoise maneuver and used to retrieve a single high-resolution BrO profile, with the standard optimal estimation mathematics (Rodgers, 2000) as described in the supplement.

Figure 3 shows the high confidence in the retrieval below the maximum flight altitude. For this profile, the aircraft flew between 64 and 967 m above ground level, and all averaging kernels in this range are close to 1. Further, the averaging kernels peak sharply at their own altitude and generally decrease by roughly 90% at the two nearest altitudes (i.e $\pm50$ m), indicating that the retrieved mixing ratio at each altitude is primarily impacted by the measurement sensitivity to that same altitude, and thus that this retrieval has a high vertical resolution, similar to the 50 m model resolution. The degrees of freedom (DOF, trace of the averaging kernel matrix) indicate how many independent pieces of information we can retrieve from each profile. For this case, we see that the 16.7 DOF correspond to highly independent retrievals below roughly 850 m (due to the 50 m grid

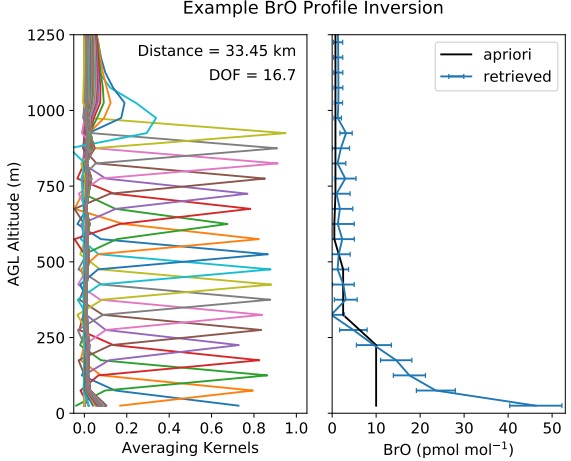

**Figure 3.** Example BrO profile retrieval from the fifth aircraft profile on March 30 with the averaging kernels on the left showing very good resolution and a high total DOF (16.7). The a priori profile is shown in black and the retrieved profile and uncertainty are shown in blue on the right. This vertical profile spanned a horizontal distance of 33.45 km.

spacing). All retrieved profiles were filtered based on confidence (DOF > 7.5), spatial resolution (horizontal porpoise distance < 100 km), and flight altitude (minimum altitude < 300 m) resulting in 245 BrO profiles from the full campaign (from February 27 to April 16 of 2022).

## 3 Results

### 3.1 Average BrO Profile

Using both sides of the aircraft profiles (down and up) results in very well-resolved BrO profiles throughout the lower atmosphere, as shown in Fig. 3, with a mean DOF of 18 throughout the campaign. This sampling also includes some profiles where only the descent was used in cases where there was no immediately proceeding ascent, or vice versa. However, these profiles often lead to good retrieval results as well, with a mean DOF of 16.

The focus of this study is to improve the understanding of reactive bromine chemistry of the springtime Arctic. Therefore, the median BrO profile from the full campaign is of particular interest. Figure 4 summarizes all BrO profiles with a box and whisker plot. The median BrO mixing ratio is shown as the center line with the boxes depicting the inter-quartile range of the observations and the $5^{th}$ and $95^{th}$ percentile intervals as whiskers at each altitude.

We retrieved a median BrO mixing ratio of 29 (mean = 30, s.d. = 13) pmol mol$^{-1}$ in the lowest grid cell (from 0–50 m), and BrO quickly decreases above the Earth's surface, likely due in part to the typical thermodynamic stability of the atmosphere in the springtime Arctic that inhibits vertical mixing (Bradley et al., 1992; Creamean et al., 2021; Peterson et al., 2015). Above the surface, BrO is roughly constant near 10 pmol mol$^{-1}$ for the next 150 m, possibly due to a residual layer above the common

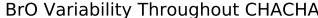

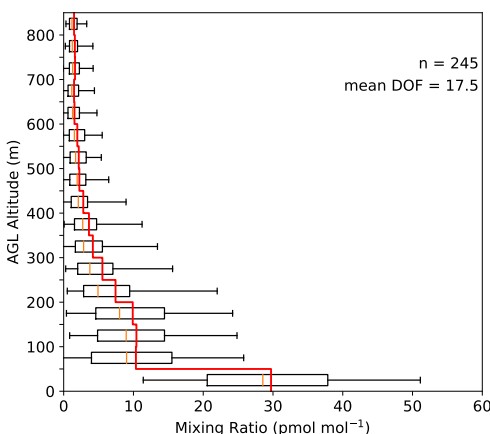

**Figure 4.** A boxplot summary of all BrO profiles from throughout the CHACHA field campaign. The boxplot indicates the median BrO mixing ratio at each altitude (orange), with the edges of the box denoting the $25^{th}$ and $75^{th}$ quantiles and the whiskers denoting the $5^{th}$ and $95^{th}$ quantiles. In red is the mean BrO box profile, which demonstrates the quick decrease of the BrO mixing ratio with altitude.

surface inversions. Above 225 m, BrO decreases to the a priori value of 1 pmol mol$^{-1}$ with mean values comparable to the standard deviation. However, it is clear from the width of both the boxes and whiskers that there is much variability in the BrO profile below 400 m altitude, where the inter-quantile range of the observations is greater than 3 pmol mol$^{-1}$ and greater than 15 pmol mol$^{-1}$ at the Earth's surface.

### 3.2 BrO Profile Uncertainty

The uncertainty of the retrieved BrO profiles can be attributed to a combination of smoothing error, retrieval noise, and forward model parameter error (each described in the supplement, Rodgers, 2000). The average BrO profile uncertainty peaks at the surface with a value of 5 pmol mol$^{-1}$. This is in part due to the fact that viewing angle uncertainty and particle extinction uncertainty have a larger impact on BAMF calculations near the surface, where light path truncation by the surface is greatly affected by both parameters. The BrO profile uncertainty decreases with altitude to be less than 3 pmol mol$^{-1}$ above 250 m, at which point it linearly decreases to roughly 1 pmol mol$^{-1}$ above 1 km altitude. The mean BrO profile error is roughly equal to the mean BrO profile above 500 m altitude. Similarly, the standard deviation of the retrieved BrO profiles is larger than the mean uncertainty below 450 m, and is over twice as large at the surface. Therefore, most of the variance observed in the BrO profiles in the lower atmosphere is significant.

### 3.3 Cluster Classification of the BrO Profiles

Previous BrO MAX-DOAS studies investigate BrO profile shapes through calculations of lower tropospheric vertical column densities (LT-VCD) and f200, where the LT-VCD is the vertical BrO column density from 0-2 km and f200 is the fraction of

the LT-VCD below 200 m (Peterson et al., 2015; Simpson et al., 2017). Due to the aircraft measurements and the resulting increased DOF, we are able to study differences in BrO profile shapes with finer detail.

To characterize the large vertical variability of BrO, we used a K-means cluster analysis of the BrO profiles (MacQueen, 1965; Lloyd, 1982). A cluster analysis is an unsupervised algorithm that simply finds common BrO profile shapes that occurred throughout the campaign, and there are several tests to determine how many clusters to include. We utilized a silhouette score (Rousseeuw, 1987) that quantifies (on a scale from $-1$ to 1) how much better a profile's cluster explains its shape compared to the next closest cluster. The highest average silhouette score corresponded to two clusters. However, this resulted in many clusters with negative silhouette scores, indicating that they may be assigned to the wrong cluster. We therefore opted for the second highest average silhouette score (Fig. S4), which also minimized negative silhouette score profiles, and corresponded to four BrO clusters. The utility of the silhouette scores is that it ensures no two clusters are too alike. If that were the case, the algorithm would suggest the number of cluster be reduced and the two clusters would be merged together.

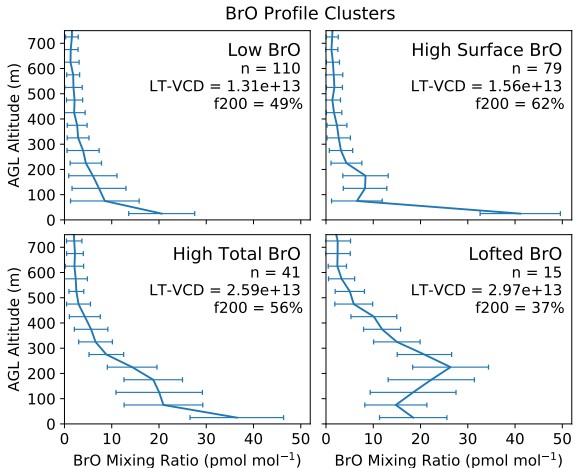

**Figure 5.** The four BrO profile clusters identified from the entire dataset. Profile clusters are shown in decreasing occurrence with low BrO being the most common and lofted BrO the least common. Error bars denote the standard deviation of all clustered profiles around the mean profile. Each plot shows the number of observed profiles in each cluster along with the mean lower tropospheric (<2 km) vertical column density and the ratio of the LT-VCD below 200 m.

The four identified clusters are shown in Fig. 5, which are defined from most common to least common as: low BrO, high surface BrO, high total BrO, and lofted BrO. The low BrO case was identified 44% of the time. Although there was still 20 pmol mol$^{-1}$ of BrO at the Earth's surface (between 0 and 50 m), there was very little (<10 pmol mol$^{-1}$) above the surface on average. Similarly, the high surface BrO case (33% of the retrievals) also resulted in less than 10 pmol mol$^{-1}$ above the surface, although there was roughly twice as much BrO at the surface compared to the low BrO case. The high total BrO case was identified 17% of the time and resulted in large observed BrO vertical columns due to a combination of high BrO at the

surface and aloft. Least frequently (6% of the retrievals), we observed lofted BrO profiles, with 73% of these 15 observations occurring on a single day (March 19, 2022), which yielded in the largest observed BrO vertical columns.

The LT-VCD and f200 observed here (Fig. 5) are consistent with previous literature. Simpson et al. (2017) also observed daytime surface BrO mixing ratios often ranging from 10–40 pmol mol$^{-1}$ near Utqiaġvik in spring of 2012. For a subset of this same data from roughly March 18–24, 2012, Peterson et al. (2015) regularly observed daytime BrO mixing ratios of roughly 15 pmol mol$^{-1}$ with a low LT-VCD (generally less than 2x10$^{13}$ molec./cm$^2$) and a f200 of roughly 50%. This closely resembles the statistics of the low BrO cluster. Peterson et al. (2015) also showed little relationship between surface BrO concentrations and LT-VCDs, which was also evident here where a doubling of the surface mixing ratio between the low BrO and high surface BrO clusters led to only a small increase in the LT-VCD. Also in agreement with Peterson et al. (2015) is the fact that the highest LT-VCDs were observed when BrO was extended to higher altitudes (200-400 m) in the lower troposphere (lofted BrO cluster).

## 4  Discussion

### 4.1  Sampling Bias Considerations

It is important to note that the findings of this study have a clear-sky bias, because ALAR could only fly in the absence of clouds to avoid icing, and the porpoising maneuver requires good visibility near the ground. This selection bias is made clear by the results of the aerosol particle inversion from each flight. Throughout the campaign, the mean 361 nm AOD retrieved from the O$_4$ inversion was 0.13 ($\pm$ 0.05). On average, the highest particle extinction was found at the Earth's surface (0.3 km$^{-1}$$\pm$ 0.4 km$^{-1}$) and quickly decreased with altitude. Above 450 m, the average particle extinction was relatively constant at 0.05 km$^{-1}$ ($\pm$ 0.04 km$^{-1}$) until roughly 1.5 km altitude.

These observations appear to be typical for this time and location based on data from the Aeronet station in Utqiaġvik. Aeronet observations (Holben et al., 2001) from March and April from 2010 to 2022 showed an average 340 nm AOD of 0.15 with a standard deviation of 0.06, which is very similar to the campaign average retrieved from HAIDI. A similar particle extinction profile shape was observed in 2020 during MOSAIC (Ansmann et al., 2023) with the highest springtime extinction at the surface and decreasing with altitude, albeit for a different Arctic location (>85°N, northeast of Greenland).

Our clear-sky sampling bias becomes apparent when considering observations from a total sky imager at the ARM facility in Utqiaġvik which observes cloud fraction throughout the year, often with 30 s resolution (Flynn and Morris, 2015). For daytime observations during March and April from 2010-2022, 56.4% of observations had less than 20% cloud cover, and this was slightly higher in 2022 at 62.9% of observations. However, for flight days during CHACHA, 81.6% of observations had less than 20% cloud cover. There were also much fewer overcast days in the CHACHA flight day subset. So although clear-sky conditions were relatively common in Utqiaġvik, they are still over-represented in this dataset. Aeronet data from 2010-2022 shows 86.3% of days with a 340 nm AOD of less than 0.2. Although, as Aeronet is a direct-sun radiometer, these results may also have a clear-sky bias.

Despite the clear-sky bias of this study, several research flights occurred on days preceding and following blowing snow events. So although no HAIDI observations directly probed halogen chemistry under such conditions (Jones et al., 2009), our observations should have some connection to these conditions that can be investigated in the future.

## 4.2   Profiles Indicate Snowpack as an Important Source of Reactive Bromine

Throughout the campaign, we often retrieved the highest BrO mixing ratios at the Earth's surface. This near-snowpack peak is
clear in the median retrieved profile (Fig. 4) and in the clustered profile shapes (Fig. 5) where 94% of the profiles fit into the first three clusters that have the highest BrO mixing ratios at the snowpack surface. This profile shape underscores the importance of multiphase snowpack surface chemistry on reactive bromine in the springtime Arctic, as well as the atmospheric stability of the area during this time of year. The high mixing ratios at the Earth's surface likely occur due to numerous recycling pathways of different reservoir species that produce $Br_2$ on the snow surface (Pratt et al., 2013; Custard et al., 2017; Halfacre et al.,
2019; Wang et al., 2019). The reactive bromine is then trapped near the ground by atmospheric stability (Bradley et al., 1992; Peterson et al., 2015).

However, BrO mixing ratios were not always highest at the Earth's surface, as shown by the lofted BrO cluster (Fig. 5). HAIDI observations were largely made under conditions with little cloud cover, no inclement weather, and good visibility near the ground (Sect. 4.1). For one of the few days (March 19) where we observed a small amount of haze (AOD = 0.21), we
retrieved multiple lofted BrO profiles.

## 4.3   Lofted BrO Profiles on March 19, 2022

The cluster analysis revealed that 73% of the lofted BrO cases occurred on a single day (March 19, 2022 shown in Fig. 6). This day was noted as being hazier than a typical flight day (visibly seen in Fig. S6), and our particle extinction inversion supports that (Fig. 7). The 361 nm AOD retrieved for this flight ($0.21 \pm 0.03$) was the second highest of the campaign and more than
a standard deviation higher than the campaign average (0.13, standard deviation of 0.05). The largest particle extinction for this day was retrieved at 225 m above the ground, and the average of the retrieved BrO profiles for this day also peaked at the same altitude with a maximum mixing ratio of 25 pmol mol$^{-1}$. This vertical correlation could signal that the BrO observed on this day may have advected from the Chukchi Sea to the west and been maintained through multiphase chemistry on airborne particles. Alternatively, this relationship could also be caused by increased mixing from the Earth's surface, lofting particles
several hundred meters above the ground.

For a single profile on this day, we also observed over 10 pmol mol$^{-1}$ of BrO nearly 1 km above the ground associated with higher particle extinction at this same level, which could be due to stratified layers of the atmosphere carrying advected particles from which reactive bromine is activated on those surfaces. However, high mixing ratios of BrO lofted this far above the ground were rarely observed in the retrieved profiles.
Lofted BrO has been observed in several prior studies (Peterson et al., 2017; Simpson et al., 2017; Frieß et al., 2022). Many studies since Frieß et al. (2011) have suggested that bromine recycling could occur on aerosol particles based on an observed correlation between BrO and aerosol particle extinction, or particle surface area concentration in the case of Peterson et al.

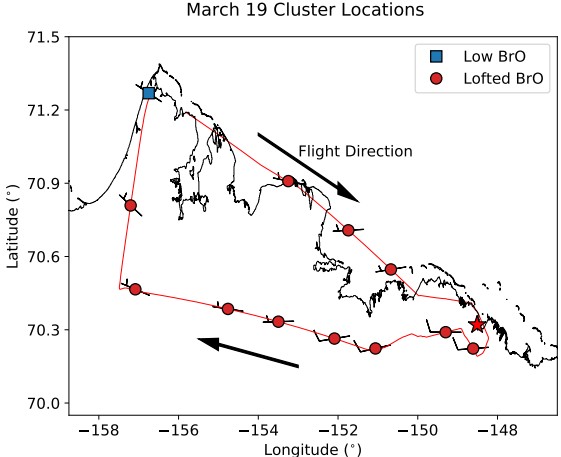

**Figure 6.** Identified BrO profile clusters for the research flight on March 19, 2022. The flight path is shown in red, beginning in Utqiaġvik and heading east along the coast of the Beaufort Sea before turning to the west around Prudhoe Bay (indicated with a star) and returning back over the tundra. The markers indicate the identified cluster and are plotted at the location of the aircraft profile, with the mean wind speed and direction from ALAR meteorological data denoted with the wind barbs. Lofted BrO profiles were identified everywhere throughout the flight, except at Utqiaġvik.

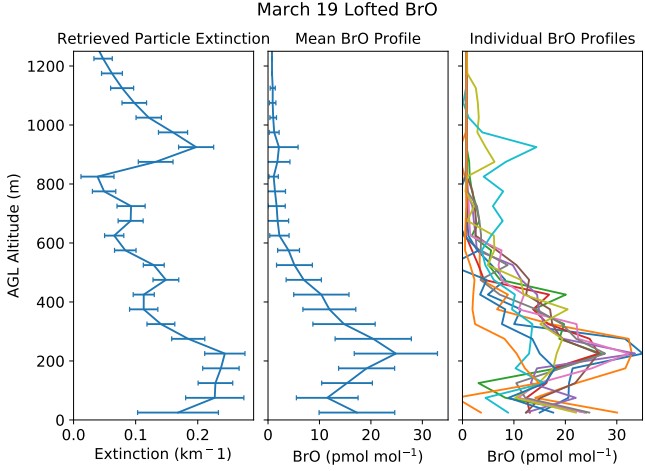

**Figure 7.** Aerosol particle and BrO inversion results from flight on March 19, 2022. The 361 nm particle extinction profile (left) shows the highest extinction at 225 m above the ground, and the mean retrieved BrO profile (middle) peaks at the same altitude. Individual BrO profiles from this day (right) indicate some profiles with BrO/reactive bromine lofted nearly 1 km above the ground. Error bars for the particle extinction are from the inversion uncertainty; error bars for the BrO profile are the standard deviation of the retrieved BrO profiles from this day.

(2017) during March, 2012 near Utqiaġvik. However, Peterson et al. (2017) also observed a distinct BrO plume nearly 1 km above the ground and clearly distinct from the boundary layer, where the lofted BrO observed here was often much closer to the ground (Fig. 7).

## 4.4 Spatial/Surface Dependence of BrO Profiles

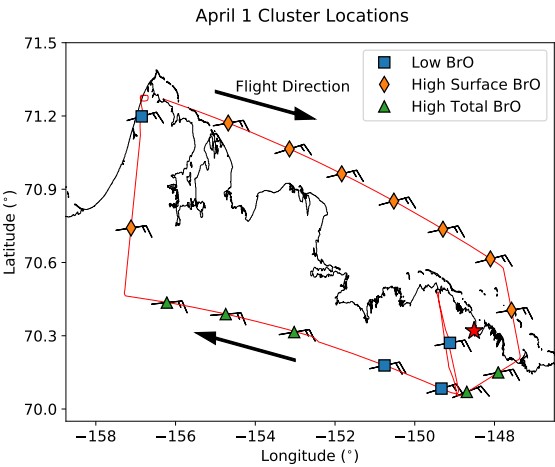

**Figure 8.** Identified BrO profile clusters for a research flight on April 1, 2022. The flight path is shown in red, beginning in Utqiaġvik and heading east over the Beaufort Sea before turning south and returning back west over the tundra. The markers indicate the identified cluster and are plotted at the location of the aircraft profile, with the mean wind speed and direction for each porpoise from ALAR meteorological data denoted with the wind barbs. The star indicates the location of the two largest $CO_2$ emitters in the Prudhoe Bay Oil Field (from EPA Greenhouse Gas Reporting Program, https://ghgdata.epa.gov/ghgp).

The four BrO clusters allow us to identify spatial changes in BrO profiles, and this is apparent for the flight on April 1, 2022 (Fig. 8). For this flight, ALAR first flew east over the sea ice out of Utqiaġvik, turned south upwind of Prudhoe Bay, then turned back to the west with two transects downwind of Prudhoe Bay before flying over the tundra while returning to Utqiaġvik. Throughout this flight, the aircraft flew vertical profiles of the lower atmosphere (e.g., Fig. 1). Figure 8 shows the flight path for this day, with the location of the markers indicating the location of the aircraft profiles and with the markers indicating the cluster into which the retrieved profile fit. On this day, there was a clear north-south gradient where observations over sea ice were exclusively high surface BrO cases and observations over tundra were often high total BrO cases. This finding could be explained by a slightly deeper boundary layer over the tundra with higher wind speeds near the Earth's surface (Fig. 9).

The exception to this clear spatial gradient on April 1 is the area directly downwind of the Prudhoe Bay region (indicated with a star on Fig. 8). For these three observations, we identified low BrO cases, likely due to direct interaction with fresh

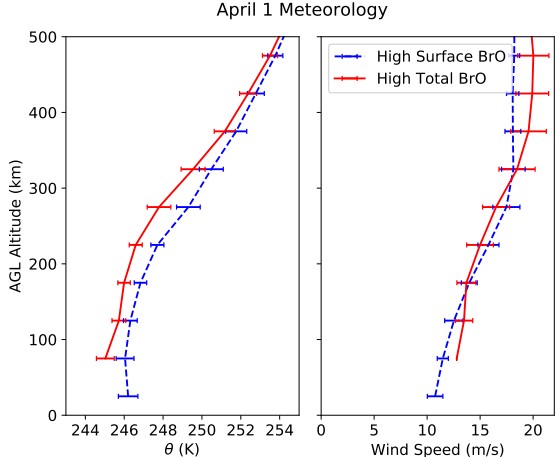

**Figure 9.** April $1^{st}$ meteorology from in-situ instrumentation on ALAR showing the average potential temperature for the high surface BrO cases (blue) over the sea ice and the high total BrO cases (red) over the tundra. The average wind speed for these porpoises is shown on the right. The lower minimum porpoise altitude over sea-ice explains the different minimum altitude of the two cases. Error bars indicate the standard deviations of the parameters.

$NO_x$ emissions from the Prudhoe Bay area that was also observed by Custard et al. (2015) and in agreement with the results of Wang and Pratt (2017), who found an impact on BrO with just 100 pmol mol$^{-1}$ of $NO_x$.

$NO_2$ profiles were retrieved using the same method and same spectra as the BrO profiles to identify BrO profiles impacted by $NO_x$ pollution. Throughout the campaign, we identified 12 profiles that appear to be impacted by elevated $NO_x$, with average retrieved mixing ratios of $NO_2$ over the lowest 500 m ranging from 0.5 to 1.4 nmol mol$^{-1}$. As our detection limit for average $NO_2$ mixing ratios was roughly 0.22 nmol mol$^{-1}$, these profiles clearly contained more than background levels of $NO_2$. Of these 12 cases impacted by $NO_x$ pollution, 11 were identified as low BrO cases (of the total 110 low BrO cases).

Although each BrO cluster was observed above both sea ice and inland snowpack, it is important to consider the impact of the snowpack surface on the retrieved BrO profiles. Since most profiles peaked toward the Earth's surface, the type of surface under each profile could affect the recycling of reactive bromine and the BrO cluster that the profile fit into. To test this, all profiles were flagged as being retrieved over sea ice or inland snowpack based on the median location of the profile in relationship to the coastline. There were too few high total BrO and lofted BrO cases to be included in this analysis, as their statistics would be skewed by flight location, i.e. all but two profiles were retrieved over tundra on March 19 when most lofted BrO cases were observed. Therefore, we only consider low BrO and high surface BrO cases.

     Between the low BrO and high surface BrO clusters, and after removing the profiles impacted by $NO_x$ pollution, 87 profiles were retrieved over land and 90 were retrieved over sea ice. However, despite this roughly even sampling distribution, only 42 of 99 (42%) low BrO clusters were retrieved over sea ice while 48 of 78 (62%) high surface BrO profiles were retrieved over sea ice, meaning that high surface BrO observations were over-represented over sea ice and low BrO cases were over-represented

**Table 1.** Distribution of BrO clusters as a function of location. The surface type was determined based on the median location of each porpoise in relation to the coast line.

|  | Tundra | Sea-Ice | Total |
|---|---|---|---|
| Low BrO | 57 | 42 | 99 |
| High Surface BrO | 30 | 48 | 78 |

over tundra. This spatial difference indicates a significant (p = 0.015 based on Fisher's exact test) location difference between these two clusters that could be driven by bromide enhancement in snow above what was likely first-year sea ice just off the coast (Peterson et al., 2019).

### 4.5 Meteorological Conditions for BrO Clusters

Considering the clear differences between the four BrO profile cluster shapes in Fig. 5, it is important to consider what factors, apart from surface type, influenced these profiles. To study this, we compiled data from the other instruments on ALAR and binned them in 25 m intervals for the same porpoises used for the clustered BrO profiles. The average profiles for the normalized potential temperature (the difference of the potential temperature profile and the profile's potential temperature at 500 m) and wind speed are shown in Fig. 10 for the four BrO profile clusters. The offset potential temperature is shown to isolate the

stability of the lower atmosphere. These plots begin at 100 m altitude, as not all porpoises descended to the same altitude.

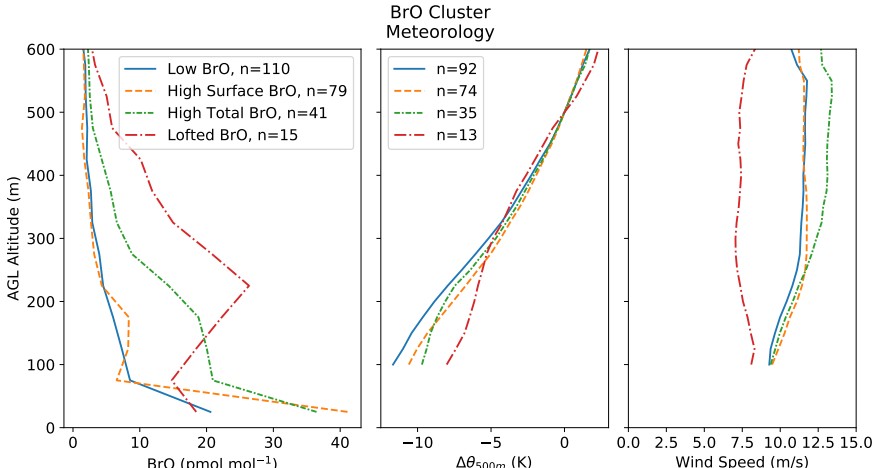

**Figure 10.** The average meteorological conditions associated with the four BrO profile clusters. The normalized potential temperature (middle, offset by the potential temperature at 500 m) shows decreasing stability associated with increasing BrO columns. The variability of these parameters is not shown here for clarity, but is indicated with error bars in the supplement (Figures S6 and S7).

The stability of the atmosphere clearly impacted the BrO profile shape, as the two most stable profiles with the largest increases in potential temperature with altitude corresponded with the low BrO and high surface BrO clusters, where there was a large difference between the BrO mixing ratio at the Earth's surface and mixing ratios aloft. The high total BrO case corresponded to a more neutral atmosphere. Similarly, the lofted BrO cases are associated with a relatively neutral atmosphere from roughly 100–300 m altitude. Because the temperature profiles did not extend to the ground, it is difficult to determine if this relates to turbulence extending from the surface or a residual layer above a stable surface layer. These lofted BrO cases are also associated with lower wind speeds throughout the lower atmosphere, which could help maintain stratification of lofted aerosol layers. Back trajectory analyses and nearby balloon-sounding data could be used to investigate these lofted profiles more closely, though this is beyond the scope of this study.

## 4.6 Associated Ozone Chemistry

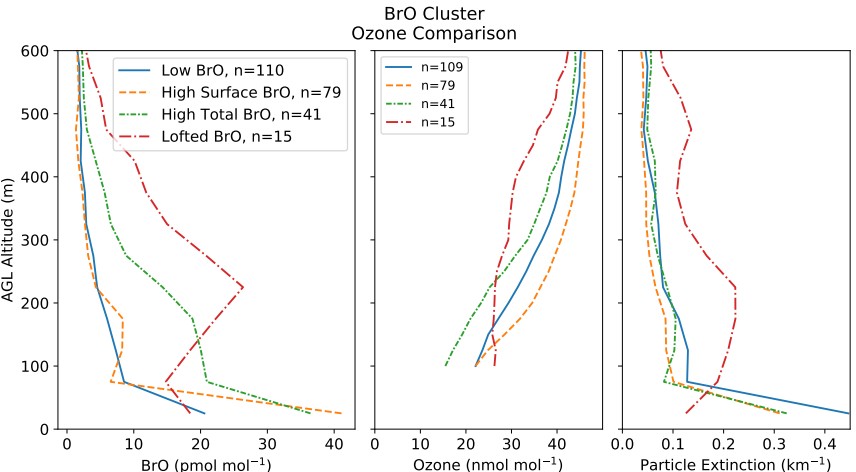

**Figure 11.** The ozone mixing ratio (middle) and 361 nm particle extinction (right) profiles associated with the four BrO profile clusters. The variability of these parameters is not shown here for clarity, but is indicated with error bars in the supplement (Figures S8 and S9).

It is also important to consider the implications of the four BrO profile clusters on ozone chemistry by studying the average ozone profiles associated with each cluster (Fig. 11). These profiles are calculated in the same manner as the meteorological parameters shown in the previous section. Also shown is the average particle extinction for each flight associated with the four clusters.

This figure reveals that there was little difference in the ozone profile above the Earth's surface for low BrO cases compared to high surface BrO cases, likely because these two BrO mixing ratio clusters are only significantly different at the surface, which was not often measured in-situ during porpoises. We also see that the lowest ozone mixing ratios are associated with the high total BrO cluster, indicating that these profiles were sampled during instances of active bromine chemistry. These three clusters correspond to very similar particle extinction profiles, which suggest few lofted particles on average during these cases

and the surface-based nature of the reactive bromine chemistry and ozone depletion for the majority of our observations. The lofted BrO cluster is associated with lower ozone mixing ratios between 200 and 500m, with a profile that reflects the increased particle extinction at higher altitudes, indicating that these may be cases where reactive bromine was activated by multiphase chemistry on the surface of the particles, impacting ozone mixing ratios aloft.

## 4.7  Satellite Implications of Surface Enhanced BrO

The frequency with which surface-peaking profiles were observed has significant implications for satellite observations for two reasons. The first is that UV-Vis satellite observations, even over bright surfaces, have lower sensitivity to trace gases at the surface than aloft (Eskes and Boersma, 2003). More importantly, the BrO mixing ratio at the Earth's surface appears to have little impact on the vertical column density. Figure 5 shows that the LT-VCD difference between the low BrO and high surface BrO clusters is very small (roughly 20%), even though their surface mixing ratios differ by a factor of 2. This similarity in BrO columns suggests that many high surface BrO conditions are likely not evident in satellite column BrO observations, which could explain some of the remaining difficulty in calculating tropospheric BrO from such measurements (Wales et al., 2023). Wales et al. (2023) also notes an underestimation of surface ozone depletion at coastal sites that could be explained by these BrO profiles. These findings point out that ODEs/reactive bromine chemistry under relatively clear sky conditions in the Arctic is a snowpack surface phenomenon and motivates the need for frequent MAX-DOAS and CIMS observations in the stable boundary layer in the Arctic to identify these high surface BrO cases.

## 4.8  Porpoising with AMAX-DOAS

The porpoising flight pattern used during CHACHA resulted in considerably higher degrees-of-freedom (DOF) of the radiative transfer inversion than ground-based MAX-DOAS instruments can achieve. The improved vertical resolution is due to the combination of observations at different altitudes, so the increased DOF is to be expected. Ground-based MAX-DOAS instruments are particularly beneficial for providing higher temporal coverage than aircraft measurements, are less impacted by adverse weather, cost considerably less to operate, and provide more vertical information than ground-based and/or in-situ observations. The combination of aircraft and ground-based MAX-DOAS observations can thus achieve both high vertical resolution and good temporal coverage. The profiles found here can be used as a priori profiles for ground-based observations to improve BrO retrievals.

Peterson et al. (2015) and Simpson et al. (2017) retrieved BrO profiles with an average of 2-3 DOF, and these works therefore reported two vertically-defined parameters to represent the information they could retrieve. It is typical for ground-based MAX-DOAS instruments to retrieve 1–3 DOF for various trace gases (Hendrick et al., 2014; Vlemmix et al., 2015; Ortega et al., 2015; Ryan et al., 2023). One way to increase DOF is by raising the MAX-DOAS instrument above the surface, which could result in upwards of 5 DOF (Koenig et al., 2017) AMAX-DOAS observations are able to retrieve considerably higher DOF, as they can sample a larger portion of the atmosphere. For instance, the work of Baidar et al. (2013) reported an average DOF of 12 for their trace gas retrievals, with lower sensitivity to the Earth's surface than found in this work due to the challenges of flying at low altitude over populated areas. Similarly, Prados-Roman et al. (2011) noted 10 DOF for BrO profiles of the entire troposphere. In

those two AMAX-DOAS studies, averaging kernels were near unity for the range of altitudes flown by the aircraft, indicating that the amount of information retrieved by the inversions could be limited by the grid spacing used. Peterson et al. (2017) utilized a finer grid spacing of 100 m and was able to retrieve up to 12.4 DOF for a profile with a maximum altitude of 1039 m (shown in the supplement of that work). Our study was able to retrieve higher DOF than other AMAX-DOAS studies due to the finer vertical resolution of the modelled atmosphere, at the cost of computational time, resulting in finer retrieved detail through the lower atmosphere. This assertion is supported by Volkamer et al. (2015) which retrieved upwards of 20 DOF for trace gas profile retrievals. This was accomplished by profiling most of the troposphere and modelling with 500 m grid spacing. The averaging kernels in our work (Fig. 3) peak at values slightly lower than 1, indicating that the 50 m grid spacing used here is roughly the highest resolution vertical spacing that we could use.

In all four AMAX-DOAS setups (Prados-Roman et al., 2011; Baidar et al., 2013; General et al., 2014; Volkamer et al., 2015), telescopes were mounted to research aircraft with some external instrumentation. This technique could be used with a less labor-intensive setup. For example, a telescope could be mounted on a window of a non-pressurized light aircraft. Similarly, drones could be used to profile the lower atmosphere with a small limb-viewing spectrometer-telescope combination. Considering the greatly improved vertical resolution of combining observations at different altitudes and the increased DOF retrieved with finer model grid spacing, this method could be expanded to quantify additional trace gases in the lower atmosphere (e.g. HCHO, CHOCHO, HONO, $NO_2$, etc.).

## 5 Conclusions

This study was aimed at better understanding reactive bromine chemistry of the springtime Arctic using aircraft BrO observations from the CHACHA field campaign based from Utqiaġvik, AK from mid-February to mid-April of 2022. We utilized the HAIDI AMAX-DOAS instrument along with a unique measurement technique of vertically profiling the atmosphere with the aircraft to retrieve dSCD profiles of BrO, $NO_2$, and $O_4$ in the lower troposphere. These observations were combined with radiative transfer model calculations to perform an optimal estimation inversion to retrieve BrO mixing ratio profiles, resulting in 245 independent, high vertical resolution BrO profiles during the campaign. BrO mixing ratios were often highest near the Earth's surface, with a mean of nearly 30 pmol $mol^{-1}$, which quickly decreased aloft. However, there was variability in the BrO profile throughout the lowest several hundred meters of the atmosphere.

A cluster analysis identified four common BrO mixing ratio profiles that occurred throughout the field campaign. A majority of the retrieved profiles revealed the highest BrO mixing ratios at the Earth's surface. These high surface mixing ratios were likely influenced by the extreme atmospheric stability of the springtime Arctic, which inhibits vertical mixing, but it also emphasizes snow over both sea ice and tundra as an important source of reactive bromine (Pratt et al., 2013; Custard et al., 2017; Peterson et al., 2018). Multiphase reactions on the surface and snowpack photochemistry (e.g., involving OH radical oxidation of $Br^-$ to produce $Br_2$, Halfacre et al., 2019) could be responsible for maintaining the higher BrO mixing ratios observed at the Earth's surface.

Similar to the 2012 BROMEX aircraft observations (Peterson et al., 2017), the least common BrO profile cluster was the lofted BrO case, where the highest mixing ratios were found above the surface. The majority of lofted BrO observations occurred on a single day, with a retrieved particle extinction profile that also peaked above the ground. These lofted profiles indicate cases where BrO was sustained through multiphase chemistry on particles that were advected from elsewhere, as found in Peterson et al. (2017), or where particles were lofted from the Earth's surface through increased turbulence. Whatever the

source of the lofted particles, this was clearly a large scale event, as these profiles were observed over a more than 24,000 km$^2$ area. Fig. S8 displays TROPOMI BrO column densities for this day, where high column values were observed over most of the observation area.

     This sampling technique involving porpoising resulted in numerous BrO profiles with considerably higher vertical resolution than previously achieved. As such, this technique should be adopted where possible. This method comes at the expense of

horizontal resolution, although that should have little impact on trace gases such as BrO that do not have obvious point sources, and the aircraft provides ready access to a variety of surface conditions. However, BrO can also be impacted by sinks with sharp spatial gradients, such as $NO_x$ plumes.

     Lastly, the resulting clustered profiles can inform, and be investigated with, 1-D chemical modelling studies based on different input scenarios. The profiles can also be implemented in MAX-DOAS studies where they can be used as the a priori

profiles. Past and future Arctic MAX-DOAS observations could be paired with these profiles to retrieve improved BrO profiles.

*Data availability.* Upon publication, data will be available at arcticdata.io

*Author contributions.* KB re-configured HAIDI for ALAR and handled instrument logistics in Germany together with DP, Airyx and the IUP Heidelberg workshop. NB, KB, and PKP calibrated HAIDI instrument. RK and PBS flew ALAR and BHS installed HAIDI and provided ALAR maintenance. KDH was mission scientist throughout campaign and provided in-situ data. NB analyzed all remote sensing data. WRS,
PBS, KAP, PKP, and TS planned each flight. NB, WRS, PKP, and KB helped develop the methodologies of this study. WRS supervised all research in this paper. NB wrote paper with input from all authors.

*Competing interests.* The authors declare that they have no conflict of interest.

*Acknowledgements.* This material is based upon work supported by the National Science Foundation under grants No. 2000403, 2000408,
2000428, 2000493, 2001449. The authors thank Raelene Wentz and UIC Sciences for logistics support throughout the field campaign as well
as the University of Heidelberg, Ulrich Platt, Denis Pöhler, Airyx GmbH, and the workshop of the IUP Heidelberg for adjusting, supplying, and assisting with the HAIDI instrument. We also thank FAA air traffic control in Utqiaġvik for their guidance throughout the field campaign along with the Jonathan Amy Facility for Chemical Instrumentation at Purdue University for their housing and maintenance of ALAR.

Aeronet and total sky imager data was retrieved at the U.S. DOE Atmospheric Radiation Measurement (ARM) North Slope Alaska site. We thank Andreas Richter for helpful discussions, providing near-real-time BrO during the field study, and providing supplemental Fig. S8, which shows the wide-spread nature of the BrO event on March 19. We lastly want to acknowledge the contribution of Sara Lance who was a PI on the field campaign and aided in flight planning.

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
