# Peer review of "HAIDI Instrument Details and DOAS Spectral Analysis"

_EGUsphere, 2023_

## Author Comment (AC1)

We thank all reviewers for their input, which we feel has clarified aspects of the manuscript and improved it. As we edited the text, we noted a few other areas where clarity could be increased. For instance, there were some discrepancies in significant digits that has been addressed.

A note to anonymous referees, any line numbers mentioned in the responses correspond to the track changes documents.

*Review #1*

*Summary*

*In their work on vertical profiles Bromine Monoxide (BrO) in the Arctic, the authors nicely motivate their investigation, put it in context of ongoing scientific discussion and – aside from presenting an excellent data set – clearly point out the scientific novelties of their work: By adapting the new flight pattern of "porpoising" to AMAX-DOAS measurements and performing the radiative transfer simulations on a finer grid, the vertical resolution of BrO profiles is improved. The higher resolved profiles are categorized into four clusters with each being investigated on chemical and meteorological effects. While the finding of high BrO concentrations close to the surface is often reported in literature, the authors used the higher vertical resolution to identify layers of increased BrO just above the surface layer. In general, the presented paper is of "outstanding" quality. In the following suggested minor revisions and technical corrections are listed:*

*Minor revisions*

*Line 238: The authors calculate box air mass factors for 4 forward viewing angles. However, these angles are not constant and depend on variations of the pitch angle on short time scales. This becomes visible in Figure S2 where the BrO DSCD peaks during the ascent when the flight altitude becomes less steep (just before 16:24), i.e. the pitch angle is smaller. The authors should include a small discussion on the variation of the pitch angle and how it can affect the retrieved BrO profiles.*

The pitch angle is accounted for by adding the average pitch angle from the 3 second integration to the relative viewing angle for each BAMF calculation. Line 249 of the track changes document now says "… by adding the mean aircraft pitch angle to the airplane-fixed viewing angles for each individual observation."

*Line 468: As the lofted BrO cluster is "clearly a large-scale event", I wonder if it could be compared to satellite retrievals. As the authors speak about this work being the link between ground based and satellite-borne measurements, a small section on satellite comparison for this exceptional case on March 19th would further prove the arguments made in this study.*

The figure below has been added to the supplement (section: March 19, 2022 Lofted BrO Profiles) showing TROPOMI BrO columns from 3-19 (via personal communication with Andreas Richter). High column values were observed over much of the region of interest,

consistent with the thought that this was a large-scale lofted event. Lines 520-521 of the main text point to this figure. A more quantitative comparison is beyond the scope of the current work.

[Figure]

Figure S8 (from personal communication with Andreas Richter).

***Technical corrections***

*Line 31: "should be used as prior profiles" – "should be used as a priori profiles"*

This key point was removed at the request of Anonymous Referee #3.

*Line 130: "the Purdue ALAR aircraft and a University of Wyoming King-Air aircraft" – Maybe swap the description of both aircrafts to get a nice transition to the next sentence.*

This change has been made in the text (line 141).

*Line 144/145 Suggestion to rephrase: "NOx emissions in the area were dominated by two specific facilities during the campaign as observed with the HAIDI nadir spectrometer, and these facilities were located very close to each other (<1 km). For the purposes of this work, Prudhoe Bay will be shown as a point source centred between these two facilities."*

*to*

*"NOx emissions in the area were dominated by two specific facilities during the campaign as observed with the HAIDI nadir spectrometer. As these facilities are in close proximity with less*

*than 1km apart, Prudhoe Bay will be shown as a single point source centred between these facilities throughout this study."*

This change has been applied to the manuscript (lines 156-158).

*Line 145: "two specific facilities" – Is there a reason as to why the name of the facilities is not mentioned here?*

The text now specifies the two facilities as the Central Compressor Plant and the Central Gas Facility (line 156).

*Line 154: "field of view of 2.8°" – Is this the FWHM or was this value calculated from the optical properties of the lens?*

The text now specifies at lines 168-169, "3.3° full width at half maximum as measured in the field, though the most upward and downward views had some truncation at the edges."

*Line 190/191: "since dSCDs are relative and SCDs depend both on stratospheric trace gas concentrations as well as solar/measurement geometry, which is observation-dependent." – I don't think this is a fair comparison of a column vs. a height resolved quantity. Also, the later introduced lower troposphere vertical column density (LT-VCD) has the same advantages as the mixing ratios. I don't think the use of mixing ratios needs to be motivated here as it is a height resolved quantity and thereby conveys more information than a column quantity like SCD or LT-VCD.*

This first sentence of section 2.4 has been removed from the manuscript.

*Line 212: "function of particle extinction" – As measurements were conducted in a cloud-free atmosphere, I'd specify this to "a function of aerosol particle extinction". This sentence could be rephrased to "The observed O4 dSCDs are reliant on the vertical profile of O4 concentration and how light travels through the atmosphere. In a cloud free atmosphere this is mainly a function of aerosol particle extinction, so these dSCD observations can be used to retrieve the vertical profile of the aerosol extinction coefficient."*

This change has been made in the text (line 226-228).

*Line 217: Why do you need to run the radiative transfer model with/without O4 absorption? Should this not be "with and without aerosol particles"?*

This is explained by Equation 7 of the supplement (line 98). The math depends on how the $O_4$ SCD is impacted by aerosol extinction, and the SCD is calculated by running VLIDORT with and without $O_4$ absorption. The algorithm therefore depends on particle extinction Jacobians from these two model calculations. A reference to equation 7 of the supplement has been added to the text at line 234.

*Line 219: "200 observations" – How many observations were there in total?*

Line 236 now specifies this is "… roughly 5% of an average 4,141 observations per flight."

*Line 220/221: Maybe it's helpful to include the information that on clear-sky days, a temperature inversion causes a stable layered atmosphere where the particle extinction profile is rather constant.*

Lines 238-239 now states "…as the mostly clear-sky nature of the days when ALAR flew is associated with strong surface temperature inversions that result in a stable atmosphere."

*Figure 2 description: Did the authors perform a sensitivity test on how the results depend on different apriori assumptions? Since it is shown later that the BrO profile correlates with enhanced aerosol extinction, I wonder if there could be an auto-correlation for cases with elevated aerosol layers.*

For a quick sensitivity test, data from the lofted BrO day on March 19 were reanalyzed with the particle extinction a priori profile halved and doubled. In both cases, the root mean square difference below 1.5 km altitude between the resulting extinction profiles and the profile used in this work was roughly 0.04 km$^{-1}$ with a mean absolute difference of 34%. This is comparable to the mean relative uncertainty of 27% produced from the original inversion and well below the 50% uncertainty assumed for the BAMF sensitivity studies now discussed in the supplement (line _). Lastly, the a priori used on this day had a maximum extinction at the surface, but each of the three retrievals resulted in the maximum extinction being located above the surface with good correlation to the originally retrieved profile (mean $R^2$ of 0.76). The results of this test have been added to the supplement (lines 127-132).

For the lofted BrO case on March 19, we are not concerned with auto-correlation, as the dSCDs reveal high BrO values aloft. The figure below has been added to the supplement (Section: March 19, 2022 Lofted BrO Profiles) and shows dSCDs from a single viewing angle for all descents/ascents from all porpoises on March 19. The specific viewing angles were chosen so that the mean viewing angle was as near-limb as possible. As can be seen, the dSCDs are high throughout much of the lowest several hundred meters.

[Figure]

Figure S7

An update of Fig. 2 in the main text now shows the impact of Rayleigh extinction along with Mie extinction, and the two are comparable throughout the lower atmosphere for an average particle extinction day. Therefore, heightened particle extinction will not necessarily have a dominant impact on BAMF calculations.

[Figure]

This figure shows two BAMF calculations for very similar measurement geometries on a lofted aerosol (March 19) day and an average aerosol (March 29) day. BAMFs are generally lower on March 19 due to an optically thicker atmosphere with higher particle extinction. However, the change is likely not drastic enough to create a lofted BrO profile without large BrO dSCDs. Further, the shape of the BAMF profiles is similar, indicating that the larger impact is likely on the magnitude of the retrieved BrO profile than on the shape.

*Figure 4: If the y-axis ticks would be at 0, 50, 100, 150, … it would be easier to identify which data points are mentioned in the text. If the result of the profile retrieval is a box-profile, maybe the box-whisker could be combined with a box-profile depiction, further enhancing the understanding of this plot. See sketched example below (green line added)*

[Figure]

This figure has been adapted in the manuscript. The new figure is attached below, with the mean BrO plotted as a box profile.

[Figure]

Figure 4

Figure 5: Is it possible to introduce LT-VCD and f200 before this figure appears?

These terms are now defined on line 296, before Fig. 5.

Line 336: "from those surfaces" – "on those surfaces"

This change has been made in the text (line 380).

Line 370 to 374: A small table of plot depicting the distribution of values would help to follow the argumentation in these two sentences.

A table has been put in the text below line 413.

Figure 10: Maybe it is helpful to add the number of cases in each profile cluster – while these numbers should be clear at this point it is quite helpful for understanding the significance of each cluster.

The number of observations has been added to this figure and Fig. 11.

Line 384: The authors should add how to identify the "two most stable potential temperature profiles" to make this sentence easier to understand.

The manuscript now discusses the "profiles with the largest increases in potential temperature with altitude" at lines 428-429.

*Line 405: I don't find this part convincing - the lofted BrO cluster is difficult to compare here as it might just show completely different airmasses near ground and above. The Ozone could also just be depleted by other processes or on the particles itself (e.g. by speeding up the sink-term reactions) and thus not be linked to bromine chemistry.*

Line 449 now specifies that "these may be cases where reactive bromine was activated by multiphase chemistry on the surface of the particles."

---

## Author Comment (AC2)

We thank all reviewers for their input, which we feel has clarified aspects of the manuscript and improved it. As we edited the text, we noted a few other areas where clarity could be increased. For instance, there were some discrepancies in significant digits that has been addressed.

A note to anonymous referees, any line numbers mentioned in the responses correspond to the track changes documents.

*Review #2*

*Brockway and coauthors present a large set of AMAX-DOAS profiles of BrO leveraging a "porpoising" flight pattern to extensively sample the surface to 1 km altitude with high resolution. BrO profiles are divided into four clusters highlighting shallow near-surface BrO enhancements, more mixed boundary layers, and lofted layers of BrO associated with particles. There is a thorough and high-quality discussion of meteorological and chemical factors and the implications for satellite retrievals and modeling of Arctic BrO. The main text and supplement together indicate that the underlying BrO are likely sound, however, a more quantitative approach to uncertainty is needed to contextualize the underlying data.*

*I have classified this as a major revision because it is necessary to ground the underlying data, and it is not yet clear that all key points are supported.*

*Major revision:*

- *More quantitative information is needed to demonstrate the validity of some retrieval choices taken by the authors. These are potential sources of uncertainty and should at least be bounded.*

1. *As the authors acknowledge in the supplement that the DOAS fitting of HCHO and BrO can have a tight anti-correlation arising from the similarity of their cross-sections (e.g. (Pinardi et al., 2013)). The authors note that HCHO is not significantly detected, and thereby infer that it can be omitted from the fits. However, this inference is not necessarily valid, without further information it is possible that the high BrO signal is limiting the ability to retrieve large HCHO columns. The critical criterion is the impact of including HCHO on the BrO dSCDs. The authors should report this and compare it to the fit uncertainty and/or the BrO columns themselves to bound the impact of this choice.*

HCHO concentrations near Utqiagvik in springtime are generally low, due in part to the lack of emissions (biogenic and anthropogenic) and the abundance of gas-phase halogens. Barret et al., 2011 observed a maximum of 200 ppt HCHO at the surface in Utqiagvik on a single day that corresponded with an ozone depletion event. To determine the impact of this much HCHO on our BrO observations, we added synthetic HCHO absorption to two different measurement spectra, one at low altitude and one at high altitude. HCHO SCDs were determined assuming well-mixed HCHO in the lowest 2 km with the modelled BAMFs and scaling the absorption cross-section to the calculated SCDs. Spectra were then fit with the original fit retrieval. In both scenarios, the maximum HCHO SCD added was roughly $1\times10^{16}$ molecules/cm$^2$ which corresponded to $1.37\times10^{13}$ and $1.63\times10^{13}$ molecules/cm$^2$ for the low altitude and high altitude

case, both less than the 1σ fit uncertainty. This is also much lower than the typical near-limb BrO dSCDs near the surface, which are on the order of $2.5 \times 10^{14}$ molecules/cm$^2$. This has been summarized in lines 14-15 of the supplement.

Adding HCHO to the fit algorithm had the impact of increasing BrO dSCDs. For one flight (April 1, 2022) where we observed high BrO and flew through a NOx plume, we ran the fit routine with a HCHO reference. This resulted in an increased BrO fit uncertainty from $2.68 \times 10^{13}$ to $2.88 \times 10^{13}$ molecules/cm$^2$ and mean increase in the BrO dSCD of $1.7 \times 10^{13}$ molecules/cm$^2$ due to mostly negative fit coefficients applied to the HCHO reference (on the order of $-1 \times 10^{16}$ molecules/cm$^2$). The change was therefore well below the detection limit of BrO. This point has been summarized on lines 15-17 of the supplement.

The negative HCHO fit coefficients are also observed in areas of high NOx where we might have expected collocated HCHO emitted from fossil fuel processing. Further, in these NOx plumes where we might expect to observe HCHO, BrO dSCDs are drastically reduced even without including HCHO in the fit routine, likely due to a direct gas phase reaction of BrO and NO$_2$.

Overall, the impact appears to be small and much less than other uncertainties in this work. An abridged version of this information has been added to lines 14-18 of the supplement.

> 2. *The authors choose to use a single constant stratospheric BrO profile for each flight. The authors note that flights are near local noon when stratospheric BrO is roughly constant. However, the springtime Arctic is expected to have high SZA (I estimate ~70° for most data from the information given but possibly more near takeoff) and therefore there might be a strong leverage on this. The authors should bound the possible impact of this choice or else provide more detail in the supplement about whether high SZA data were filtered, or what the range of SZA sampled to support their assertion.*

The median SZA of the observations during this field campaign was 70°, and the mean change in SZA during a flight was 6°. Due to the path length enhancement in the lower atmosphere of our near-limb observations, we chose not to account for diurnal variation in the stratospheric BrO profile, and since we are solving for the concentration profile near the surface, no diurnal variation needed to be applied in the lower atmosphere. Early in the campaign, some observations were made at SZAs greater than 80° (4.3% of all observations, with a maximum observation SZA of 82.2°) and these observations are not filtered out. To determine the impact of diurnal variation, we used the Theys et al. (2009) climatology with the minimum and maximum observed SZA for each flight and calculated a roughly constant 10% (1.35 ppt at the peak) uncertainty throughout the stratosphere. In this study, we propagated a 33% uncertainty in the BrO profile due in large part to the satellite stratospheric NO$_2$ observations. As a result, the uncertainty caused by diurnal variation is well within the assumed uncertainty of these profiles. This has been added to the supplement section "BrO Error Propagation and Sensitivity Studies" (lines 170-176).

> 3. *The authors have employed a random sampling of the flight to retrieve a single aerosol profile for each flight. Statistically, this approach minimizes the bias of the average of*

*the random sample. Some information to assess the success and/or validity of this approach for individual profiles is needed. Could the authors provide statistics or even better an example graph comparing measured and modeled $O_4$ dSCDs? As above, can the authors provide an estimate of the uncertainty arising from imperfect aerosol retrieval, or is this already propagated in the BrO optimal estimation? If it is the latter that is not clear. The recently published $O_4$ cross section (Finkenzeller and Volkamer, 2022) has revised the bands included for the fits here compared to the prior cross-section used (Thalman and Volkamer, 2013) would this impact the results?*

As stated in Finkenzeller and Volkamer, 2022, the novelty of their work is the extension of the $O_4$ cross-section to lower wavelengths, and the peak absorption centered around 361 nm is largely unchanged. However, the absorption band centered near 341 nm differs more significantly between the two works, and this is in the fit range used for the BrO and $O_4$ retrieval. Applying the 263 K $O_4$ absorption cross-section from Finkenzeller and Volkamer, 2022 to an entire flight (from April 1) resulted in no appreciable difference in $O_4$ or BrO dSCDs. The root mean square difference between the $O_4$ dSCDs with the different cross-sections is $1.7 \times 10^{41}$ molecules$^2$/cm$^5$, with an $R^2$ near unity and a fit slope of 1.006. BrO dSCDs are similarly unaffected with a root mean square difference of $3.9 \times 10^{12}$ molecules/cm$^2$ and a fit slope of 0.978. For both dSCDs, the fit error is slightly reduced (<2%) with the newer $O_4$ cross-section. We are therefore confident that a full reanalysis is not needed, and the differences seen in both $O_4$ and BrO dSCDs are well within the dSCD uncertainties attributed to the cross-sections (5% for $O_4$ and 9% for BrO). The supplement now states that Finkenzeller and Volkamer (2022) should be used in the future (lines 21-23).

To discuss the validity of the $O_4$ inversion, Figure S3 has been added to the supplement showing the observed vs modelled $O_4$ for the 200 data point (training) subset of each flight and for all porpoises throughout the campaign. To calculate the modelled dSCD for all porpoises, the BAMFs were used with the a priori $O_2$ profile (based on temperature and pressure), whereas the Lambert-Beer Law is used for the training subsets. This linearization could lead to some uncertainty between the two calculations. However, a larger impact arises from the fact that the training subset is modelled at 361 nm at a major $O_4$ absorption peak while the BAMFs are modelled at 350 nm for BrO. To account for this difference, the dSCDs modelled using 350 nm BAMFs were fit against the training 361 nm modelled dSCDs for each flight individually. The relationship between the two modelled dSCDs was largely linear, with the 350 nm dSCDs generally higher for low $O_4$ absorption values and lower for high $O_4$ absorption. An orthogonal distance regression was used to convert the 350 nm dSCDs for all porpoise observations to 361 nm. An example of this dSCD conversion is shown in the figure below. The average slope for this fit from all flights was 1.108, very similar to the Rayleigh extinction factor between these two wavelengths, i.e. $(361/350)^4 = 1.132$.

[Figure]

The RMSE for the entire porpoise dataset between observed and modelled (361 nm) dSCDs is $4.6 \times 10^{42}$ molecules$^2$/cm$^5$. The following figure has been added to the supplement (line 133) to show how well the modelled $O_4$ represents the observations, and these statistics have been added to the main text (lines 244-245). The $R^2$ listed in this figure is also lower than that in the text, as the original manuscript listed the mean $R^2$ for all flights while this figure lists the $R^2$ of all training and porpoise observations together. Statistics have been updated in the main text, along with a reference to this figure (line 241).

[Figure]

Figure S3

The particle extinction inversion results in an average relative uncertainty of roughly 20%. To account for the extrapolation of the retrieved extinction profile to points outside of the trained subset, this error is doubled in any figures. Further, a 50% uncertainty was used to determine the impact of particle extinction uncertainty on BAMFs. Sensitivity studies were performed to determine the impact of several uncertainties on calculated BAMFs. The resulting BAMF uncertainty was then used to vary the BAMFs and calculate the impact on the BrO inversion to include as a forward model error. The resulting variability in the BrO profile from sensitivity tests was then added as an additional uncertainty term to all BrO profiles. A new section has been added to the supplement (BrO Error Propagation and Sensitivity Studies) to discuss these sensitivity tests and a new section has been added to the results section (3.2 BrO Profile Uncertainty) of the main text to discuss the uncertainty of the individual BrO profiles.

- *Uncertainty needs to be addressed when examining differences and variability. How does the uncertainty of individual profiles compare to the variability in the clusters, e.g. in Figs. 5 and 7? In the current manuscript, there is not a clear demonstration that the increase of BrO with altitude in cluster-4 data is significant – the best case being a comparison of 50-100 m to 200-250 m in Fig. 7. If there is not a significant increase, then what is the meaningful difference between cluster 3 and cluster 4? The variability of the retrieved profiles can obscure significant differences, but some assessment of the significance of the underlying profiles is needed. How much of the increase in BrO in lofted layers arises from a decrease in light path from aerosol, and how much is driven by constant or increasing BrO dSCDs? The significance of this result especially needs more context and support.*

The K-means clustering algorithm that groups the BrO profiles is unsupervised, and simply looks for common profile shapes to minimize differences between the members of a group and the group mean. More information on the silhouette score test used to select four clusters as the best choice has been added to the text (lines 304-309). While it is true that one cluster may not significantly differ from another cluster at a given level, taking into account the entire profile shape reveals the differences between the clusters. For example, averaging the Lofted BrO cluster and the High Total BrO cluster between 200 and 400 m results in significantly different values, even accounting for the variability of the underlying profiles. The silhouette score test ensures that no two clusters are too similar, which is why we utilized the second highest score of four clusters.

While it is true that clustering the profiles can obscure some variability of the underlying profiles, the purpose of this work is to benchmark the general BrO profiles observed during this field campaign. A section (3.2) discussing the uncertainty of the underlying BrO profiles has been added to the results. In general, the uncertainty of the BrO profiles is comparable to the cluster variability above 300 m, and is lower than the cluster variability below 300 m. For this reason, the cluster variability has been shown as the error bars on the graphs.

As for the lofted profiles on 3-19, the retrieved profiles are influenced both by high dSCDs well above the surface as well as increased particle extinction. As discussed, this day was associated with higher particle extinction. Therefore, the same BrO dSCD as observed on a more normal

day would result in a higher BrO mixing ratio. A figure discussing the dSCD profile has been added to the supplement so that readers can see the underlying data that supports the lofted BrO profiles (page 10 of the supplement).

[Figure]

Figure S7

An example BAMF calculation is shown below with comparable geometry on a high aerosol day (3-19) and mean aerosol day (4-1). The BAMF is decreased on 3-19 due to an optically thicker atmosphere. But the difference does not appear large enough to create a lofted BrO layer from low dSCD values. It is difficult to determine the overall effect of these BAMF differences on BrO profiles. But as the increased particle extinction does not greatly impact the shape of the BAMF profiles but more the values, we expect the increased particle extinction to impact the magnitude of the retrieved BrO profile more than the shape.

[Figure]

***Minor revisions:***

*Line 246-248. Grid resolution and DoF do not interact linearly in this manner. The resolution can be partly inferred from the off-diagonal terms in averaging kernels which from the example in Fig. 3 show the resolution is roughly equally valid for the full altitude range. From the examples provided in kernels which peak at lower values are also broader and that the slight loss of resolution is in the middle of the profile. This is roughly as expected for porpoising maneuvers with a rigid telescope which will vary pitch away from horizontal in the middle of the profile.*

*If this is the basis of capping Fig. 4 and other figures at 850 m, they should be extended if there are sufficient statistics at higher altitudes.*

This sentence has been removed from the text.

Figure 4 is capped at 850 m because not all porpoises rose to this level. Extending the figure further just shows more influence of a priori assumptions.

*Line 254 -257: It appears inconsistent that inhibited vertical mixing can explain high concentration below 50 m, but consistent mixing is invoked to explain near constant mixing ratios in the next 150 m. Can other hypotheses be offered, or else some discussion of variability in vertical mixing? From the cluster analysis it appears that this is a consistent feature of all clusters so I would suggest formulating a different hypothesis. Is it an effect from the 23% of data in clusters 3 and 4, or is it a result of an active source? This is partly addressed in Sects. 4.2 and 4.5, but the description should be consistent in different parts of the text. As discussed in Sect. 4.5 discussion is somewhat limited by filtering data from the lowest altitudes. Some more discussion of cluster 3 might be useful to addressing this.*

The text now states that this may be due to a residual layer above the common surface inversions (lines 277-278).

*Line 258: Above 250 m, the near-zero BrO omits some potentially important detail. Most of the retrieved DoF are above this altitude. How do the medians and ranges compare to the a priori?*

The text now states: "BrO decreases to the a priori value of 1 pmol mol$^{-1}$ with mean values comparable to the standard deviation" (lines 279-280).

*Line 414: A recent publication (Wales et al., 2023) examines satellite and model surface BrO in the Arctic. Do the authors believe the latest findings address the limitations they outline?*

This manuscript emphasizes the limitations listed in Wales et al. (2023), which discusses a case where their method was unable to reproduce large, near-surface BrO mixing ratios observed with ground-based instrumentation in Utqiagvik. Similarly, Table 4 of that study shows the lowest correlation between their modelled value and 0-200 m BrO columns observed via O-BUOY data. Our manuscript also suggests that satellite columns are likely less sensitive to surface-based differences in BrO, such as this difference between clusters 1 and 2. The reference has been changed to Wales et al. (2023) as this is a much more recent publication to calculate tropospheric BrO amounts (line 458).

*Sect. 4.8: Some discussion of deeper AMAX-DOAS profiles such as those in (Volkamer et al., 2015) and (Koenig et al., 2017) is warranted. While it focuses on CHOCHO rather than BrO, (Volkamer et al., 2015) also includes a case study comparing AMAX-DOAS, surface MAX-DOAS (shipborne), and in situ (CE-DOAS) detection which is relevant to the discussion here.*

More literature of MAX-DOAS DOFs has been added to the Section 4.8 as well as the impact of raising the MAX-DOAS platform above the surface (lines 475-477, 486-488).

***Technical comments***

*Line 151: "dimer" should not be used to describe $O_4$ for the pressures and temperatures in Earth's atmosphere. The other language used such as "associative collision" is more accurate, but it is best described as "collision induced absorption".*

This change has been made in the text at line 166.

*Line 154: The mean elevation angles are not evenly spaced, as such it seems unlikely that the field of view is the same for all angles. Can the language be clarified?*

The view is truncated at both ends, so the first and fourth elevation angle have decreased input on the extreme ends. As a result, we used the mean incoming elevation angle instead of the geometric center of each view to better represent the viewing angle of the telescopes. The reviewer is correct that they do not necessarily have the same overall field of view. The four views all have the same FWHM as calculated with field data, as now stated for clarity on lines 169-170 of the revised manuscript.

*Line 183-184: Does the horizontal distance estimate include the mean light path averaged over by the BrO measurement, the flight distance, or both?*

The manuscript has been updated to explain that this is only the flight distance (line 199).

*Line 219: Can the authors provide more detail on the random sampling? What fraction of total measurements are the randomly selected 200? Are the 200 measurements selected independently or are later selections modified to ensure coverage?*

The 200 points were selected randomly for the porpoises, which generally ensures coverage. Full coverage of the range of flight altitudes was also confirmed after model runs. The average flight consisted of 4,141 observations, so the training dataset was roughly 5% of the entire dataset. This has been included in the revised text (line 236).

*Fig. 2: The Rayleigh extinction profiles should be shown for comparison.*

This has been added to the figure. The new figure is shown below.

[Figure]

Figure 2

*Line 368: I believe e.g. should be i.e.*

This has been updated in the text (line 412).

*Fig. 11: I recommend showing ozone to zero in the middle panel.*

This change has been made to the figure.

*Line 425: "prior profiles" here should be "a priori profiles"*

This change has been made (line 470).

*Supplement Fig. S1: The authors invoke a cutoff effect from intercepting the surface to explain the positive $O_4$ optical density. How relevant is this effect over a high-albedo surface? Since BrO (and perhaps $NO_2$) also are typically maximum near the surface why is this effect relevant for $O_4$ but not BrO or $NO_2$?*

Even over a high albedo surface, light-path truncation occurs close to the surface, as the downwelling radiation is more a function of solar geometry. The reference spectra used throughout the field campaign are near-limb observations from roughly 1000 m altitude. The $O_4$ dSCDs often oscillate between positive and negative values based on the observation altitude and upward and downward aircraft pitch angles. In this case, the light path is truncated by the presence of the surface, so the observation "light path" is shorter than that of the reference spectrum. The same phenomenon occurs for the BrO and $NO_2$ observations, but the trace gas abundance at the surface is considerably larger than that aloft, leading to positive absorption relative to the reference. Clarifying information has been added to this figure caption.

References:

Barret, M., Domine, F., Houdier, S., Gallet, J.-C., Weibring, P., Walega, J., Fried, A., and Richter, D, Formaldehyde in the Alaskan Arctic snowpack: Partitioning and physical processes involved in air-snow exchanges, J. Geophys. Res., 116, D00R03, doi:10.1029/2011JD016038, 2011.

---

## Author Comment (AC3)

We thank all reviewers for their input, which we feel has clarified aspects of the manuscript and improved it. As we edited the text, we noted a few other areas where clarity could be increased. For instance, there were some discrepancies in significant digits that has been addressed.

A note to anonymous referees, any line numbers mentioned in the responses correspond to the track changes documents.

**Review #3**

*This manuscript offers new DOAS-measured BrO profiles from an aircraft, making observations at various altitudes to profile profiles at a high-altitude resolution. They observe different concentrations and profiles of BrO and report a lofted BrO profile. The paper is well written with a detailed discussion on the meteorological effect, although implications on chemistry are not explored in detail. Overall, the paper adds to the current literature but needs a few details before publication:*

*Comments:*

*Line 15: 'at the Earth's surface' is not necessary.*

We wanted to be careful to differentiate the Earth's surface from the surface of particles, which is important in this chemistry. This sentence remains unchanged to keep that clarity.

*Line 20: MAX-DOAS profile retrievals do not necessarily depend on prior BrO profiles. This depends on the method used for profile retrievals. Please remove this claim from the abstract and clarify this in the text.*

The text (lines 20-21 of the revision) now specifies that these profiles and their uncertainties will help some future studies that rely on optimal estimation inversion algorithms.

*Key point number 4: This is not a key point from the study but a future outlook – it does not belong in the key points.*

This text has been removed.

*Line 66: This is mainly driven by chlorine chemistry, with some contribution from bromine chemistry.*

This has been specified in the text (line 67).

*Line 80: Also mention how climate change is leading to increased iodine chemistry impacts (Benavent et al., 2022) along with bromine and chlorine.*

Benavent et al., 2022 and the possible role of iodine in Springtime Arctic chemistry has been added (lines 83-85).

*Line 91: Add papers (Tuckermann et al., 1997; McElroy et al., 1999; Carlson et al., 2010; Liao et al., 2011; Benavent et al., 2022; Zilker et al., 2023)*

The references have been added (lines 95-97).

*Line 93: Please cite original papers that developed profile inversions rather than a later self-cited work.*

This sentence is not meant to discuss the development of methods, rather the use of MAX-DOAS to retrieve BrO profiles in the Arctic. This reference has been changed to Frieß et al., 2011 to avoid self-citation (line 99).

*Line 96: Please cite the original work that led to the inclusion of halogens in chemistry models instead of only citing your own works, e.g. (von Glasow et al., 2002).*

The reference has been added (line 103).

*Line 103: 'the same instrument is used in this study.'*

This change has been made in the text (line 110).

*Line 105-115 –the text is dedicated to the BROMEX campaign that the authors participated in, but all the subsequent studies by other groups that have increased our understanding of bromine chemistry have been ignored.*

This paragraph is meant to motivate the CHACHA field campaign, which bears many similarities to the BROMEX field campaign from 10 years prior, hence the focus on the BROMEX campaign. More references have been added to this paragraph to reflect discoveries made outside of this field campaign (lines 120-123).

*Line 140: Few flights went much south of Atqasuk, Alaska, at which point the topography started to rise, so the ground elevation was often close to sea level – not clear how the ground level is close to sea level if the topography is rising.*

Since there were only a few observations south of Atqasuk where the ground elevation starts to rise, the ground elevation of most observations was close to sea level. The text now states, "Few flights went further south than Atqasuk, Alaska, where the topography starts to rise, so the ground elevation was close to sea level for most observations." (lines 152-153)

*DOAS settings – not including HCHO is not standard due to the substantial interference between BrO and HCHO. The authors mention that it did not have any effect, but no evidence for this is provided. Please demonstrate that the exclusion of HCHO did not affect the BrO fits and present a correlation plot between HCHO and BrO through the campaign in high and low HCHO regions.*

HCHO in the Arctic springtime is typically very low, at 100-300ppt, at the surface (Sumner et al., 2002), so HCHO was omitted from the fit algorithm due to the noted substantial interference. Adding a HCHO reference to the fit routine for the flight on April 1, 2022 leads to insignificant,

increased BrO dSCDs and almost entirely negative HCHO dSCDs. A detailed reasoning for this omission can be seen in the response to Anonymous Referee #2.

*How the authors deal with short-term variations of the aircraft pitch angle is unclear. It would be nice to see some sensitivity analysis or a discussion on the effect of short-term variations of the pitch angle.*

The average pitch angle of each observation is added to the relative viewing angles for each BAMF calculation. The mean pitch angle change between observations is 0.3°, so it is not accounted for in the radiative transfer calculations. The field of view of the telescope is much larger than this and is used as the basis for a sensitivity study on the propagation of the elevation angle uncertainty. Impacts of viewing angle uncertainty have been added to a new section of the supplement (BrO Error Propagation and Sensitivity Studies) (lines 188-192).

*Line 183: Is the horizontal distance just the flight path or includes the light path?*

The text now specifies that the horizontal distance is only from the flight path (line 199).

*It is not clear where the reference spectra were collected from – were they collected for each flight individually – how was the area with 'low' trace gas concentration determined? If not, what is the effect of this?*

Fifteen reference spectra were used throughout the campaign to account for changing solar zenith angles and stratospheric BrO throughout the campaign. These references are observations from higher-altitude (~1000 m) portions of different flights. Low trace gas concentrations were generally determined by using the lowest SZA observations at the highest flight altitudes. Though some trial and error was involved if large negative dSCDs were present. This is now stated in the supplement at line 6.

*It would also be nice to see the mean vertical ozone profiles in the lower 100 m, as bromine chemistry is highly active there. The authors have the data, why not show it?*

Ozone data was measured in-situ on the aircraft. As the bottom altitude of the porpoises varied, and only rarely approached the surface during missed approaches at nearby airports, this data is not shown as it skews the shape of the profiles based on available data. Therefore, we only show data above 100 m so that the mean at each altitude is based on a consistent dataset.

*Looking at the plots, it is unclear how the 4 clusters differ. The lofted BrO profile is indeed different, but aside from that, the other profiles are not very different when considering the variation.*

The K-means clustering algorithm used to create the 4 clustered profiles is an unsupervised algorithm that combines profiles based on how alike they are. The silhouette score test used to choose four clusters shows that three and five clusters fit the data worse than four clusters. The shapes of each of the first three profiles are similar, indicating the surface-based chemistry claimed in this manuscript. However, the magnitude and rate of decrease with altitude are clearly

different for all three cases. The difference between 20 and 40 pmol mol$^{-1}$ of BrO at the surface is significantly different from a chemistry perspective.

*The low-BrO day still has 20 pptv at the surface – does the ozone profile reflect this? The ozone profiles in the supplementary text show ozone mixing ratios only above 100 m.*

The available mean ozone profile is shown in Figure 11. Ozone data is not available at the surface for most observations. There are instances of Low BrO profiles associated with both high and low ozone concentrations, indicating that this profile can be associated with cases of low reactive bromine chemistry as well as ODEs that can result in low BrO concentrations.

*The authors should include a comparison with satellite observations, especially for the lofted BrO day, which looks like a widespread event.*

A new figure (S8) has been added to the supplement showing TROPOMI BrO observations (via personal communication with Andreas Richter) that shows high BrO columns on March, 19 over the measurement region, agreeing with the assertion that this is a large scale event. A full satellite comparison is beyond the scope of this work.

*If inhibited vertical mixing explains values close to the surface, does that mean that the surface ozone was completely depleted?*

As seen in figure 11, ozone was generally depleted towards the surface. Although $O_3$ measurements from the aircraft were not generally made below 100m, this profile shape was common throughout the campaign, consistent with inhibited vertical mixing. Previously, Peterson et al. (2016) reported full ozone profiles for several BROMEX profiles over Utqiaġvik and Atqasuk, with comparisons to ground-based data, showing that significant changes in ozone levels can occur within the lowest 100 m. Oltmans et al. (2012) previously reported ozone vertical profiles up to 2 km using ozonesonde data at Utqiaġvik.

*Why is there a cutoff effect for O4 but not BrO or NO2?*

The dSCDs of $O_4$ were often negative due to light path truncation by the surface. The same effect occurs in the observations of BrO and $NO_2$. In this observation however (from figure S1), the concentration of BrO and $NO_2$ was considerably higher at the surface than at the altitude at which the reference spectrum was recorded. Therefore, the higher concentrations overcome the shorter path length, resulting in positive absorption. This has been added to the caption of Fig. S1.

References:

Frieß, U., Sihler, H., Sander, R., Pöhler, D., Yilmaz, S., and Platt, U., The vertical distribution of BrO and aerosols in the Arctic: Measurements by active and passive differential optical absorption spectroscopy, J. Geophys. Res., 116, D00R04, doi:10.1029/2011JD015938, 2011.

Oltmans, S. J., Johnson, B. J., and Harris, J. M., Springtime boundary layer ozone depletion at Barrow, Alaska: Meteorological influence, year-to-year variation, and long-term change, J. Geophys. Res., 117, D00R18, doi:10.1029/2011JD016889, 2012.

Peterson, P. K., Pratt, K. A., Simpson, W. R., Nghiem, S. V., Pérez, L. X., Boone, E. J., Pöhler, D., Zielcke, J., General, S., Shepson, P. B., Frieß, U., Platt, U., and Stirm, B. H.: The role of open lead interactions in atmospheric ozone variability between Arctic coastal and inland sites, Elementa, 2016, https://doi.org/10.12952/journal.elementa.000109, 2016.

Sumner, A. L., Shepson, P. B., Grannas, A. M., Bottenheim, J., Anlauf, K. G., Worthy, D., Schroeder, W. H., Steffen, A., Domine, F., Perrier, S., and Houdier S.: Atmospheric chemistry of formaldehyde in the Arctic troposphere at Polar Sunrise, and the influence of the snowpack, Atmos. Environ., 36, 2553–2562, 2002.